# Evolution of high-molecular-mass hyaluronic acid is associated with subterranean lifestyle

Yang Zhao [1,2,6], Zhizhong Zheng[1,6], Zhihui Zhang[1], Yandong Xu[2], Eric Hillpot[1], Yifei S. Lin[1], Frances T. Zakusilo[1], J. Yuyang Lu[1], Julia Ablaeva [1], Seyed Ali Biashad[1], Richard A. Miller [3], Eviatar Nevo[4], Andrei Seluanov [1,5] ✉ & Vera Gorbunova [1,5] ✉

Hyaluronic acid is a major component of extracellular matrix which plays an important role in development, cellular response to injury and inflammation, cell migration, and cancer. The naked mole-rat (*Heterocephalus glaber*) contains abundant high-molecular-mass hyaluronic acid in its tissues, which contributes to this species' cancer resistance and possibly to its longevity. Here we report that abundant high-molecular-mass hyaluronic acid is found in a wide range of subterranean mammalian species, but not in phylogenetically related aboveground species. These subterranean mammalian species accumulate abundant high-molecular-mass hyaluronic acid by regulating the expression of genes involved in hyaluronic acid degradation and synthesis and contain unique mutations in these genes. The abundant high-molecular-mass hyaluronic acid may benefit the adaptation to subterranean environment by increasing skin elasticity and protecting from oxidative stress due to hypoxic conditions. Our work suggests that high-molecular-mass hyaluronic acid has evolved with subterranean lifestyle.

Subterranean species have been widely studied due to their adaptation to extreme environments, such as hypoxia, hypercapnia, and harsh conditions underground. Some subterranean species such as naked mole rats (NMRs)[1] and blind mole rats (BMRs)[2,3] display extreme longevity and resistance to cancer. Interestingly, despite NMR and BMR having convergently evolved subterranean lifestyle, both species produce abundant high-molecular-mass hyaluronic acid (HMM-HA) in their extracellular matrix (ECM). There have been several reports investigating the genomic similarities that could explain the mechanisms behind adaptations to subterranean life[4,5]. However, the role of HMM-HA in adaptation to subterranean environment remains poorly understood.

Hyaluronic acid (HA) is a linear polysaccharide and a major component of the ECM. A polymer of 2 interchanging carbohydrates, D-glucuronic acid and N-acetyl-D-glucosamine[6], HA can reach sizes upwards of 6 mega Daltons (MDa). This is achieved by the polymerization of HA at the cell membrane[7]. HA is synthesized by three hyaluronan synthases (HASs), namely HAS1, HAS2, and HAS3. These isoenzymes, all of which are transmembrane glycosyltransferases, have similar structures. HAS1 and HAS3 synthesize polymers of relatively smaller size, while HAS2 synthesizes the longer polymers[8]. The differential effects of small HA polymers (low molecular mass HA, LMM-HA) and HMM-HA have been studied. LMM-HA has been associated with inflammation[9-11], angiogenesis[12], cell migration[13], and hyperalgesia[14]. HMM-HA, on the other hand, has anti-inflammatory[15], anti-hyperalgesic[14], anti-proliferative[1], and stress-resistance[16] effects. In particular, very high molecular mass HA (vHMM-HA) found in NMRs has been shown to confer a superior protection against oxidative stress to human HMM-HA[17].

[1]Department of Biology, University of Rochester, Rochester, NY 14627, USA. [2]Department of Physiology and Department of Hepatobiliary and Pancreatic Surgery of the First Affiliated Hospital, Zhejiang University School of Medicine, Hangzhou 301158, China. [3]Department of Pathology, University of Michigan Medical School, Ann Arbor, MI 48109, USA. [4]Institute of Evolution, University of Haifa, Haifa 3498838, Israel. [5]Department of Medicine, University of Rochester School of Medicine, Rochester, NY 14627, USA. [6]These authors contributed equally: Yang Zhao, Zhizhong Zheng. ✉e-mail: andrei.seluanov@rochester.edu; vera.gorbunova@rochester.edu

Though displaying diverse material properties in different organs[18], one of the stark features of NMR HA is its large size[1,17]. The HMM-HA found in NMRs has been shown to contribute greatly to the animals' resistance to cancer and other age-related diseases[1,19]. It has been hypothesized that the NMR HMM-HA evolved to provide strong and elastic skin in adaptation to squeezing through the subterranean burrows[20]. If this hypothesis is correct, one would expect HMM-HA to be found in other subterranean mammals. Remarkably, unique amino acid changes were found in *HAS2* gene in many species of African mole rats[21]. Additionally, a recent study revealed high expression of two other genes involved in hyaluronan metabolism, *hyaluronidase-3* (*Hyal3*) and *Tnfaip6* in NMR[22], suggesting that multiple genes may contribute to more abundant HMM-HA in subterranean animals.

Here we demonstrate that HA is abundant in skin, heart, and kidneys of a wide range of subterranean mammals, but not in the phylogenetically related aboveground species. Additionally, the size of the HA in subterranean species is much larger than in aboveground species. Further analysis revealed that the differential expression of HAS2, HYAL1, and HYAL2, as well as five unique mutations on HYAL2 contribute to the accumulation of abundant HMM-HA in the subterranean species. Our results suggest that abundant HMM-HA has evolved as an adaptation to subterranean environment.

## Results

### Fibroblasts of subterranean species secrete HMM-HA

To investigate if abundant HMM-HA is an adaptation to subterranean lifestyle, we examined six phylogenetically distinct subterranean species: the BMR (*Nannospalax galili*), NMR (*Heterocephalus glaber*), Damaraland mole rat (DMR, *Fukomys damarensis*), Eastern mole (EM, *Scalopus aquaticus*), star-nosed mole (SNM, *Condylura cristata*), and Transcaucasian mole vole (TMV, *Ellobius lutescens*) (Fig. 1a). We then cultured primary skin fibroblasts from these species and collected conditioned media after incubating with confluent cells for 10 days. Viscosity of conditioned media from different species was measured and compared. Except for the TMV, for which cell lines were not available, the conditioned media of all five subterranean species was more viscous than the condition media of any other aboveground species (Fig. 1b). Treatment with hyaluronidase (HAase), which specifically digests HA, eliminated the viscosity (Fig. 1b), indicating that the viscosity of the media is a result of HA secretion.

To directly compare the size of HA secreted by the cells, HA from conditioned media was purified and analyzed by pulsed-field gel electrophoresis. Three aboveground species were chosen as controls based on their closer phylogenetic relationship with the subterranean species. House mouse was chosen as a control for BMR, guinea pig (GP) as a control for NMR and DMR, and short-tailed shrew (shrew, *Blarina brevicauda*) as a control for EM and SNM. Electrophoresis showed that the HA from all five subterranean species had a high molecular mass reaching up to 8000 kDa and above, while HA of the three aboveground controls was below 6000 kDa (Fig. 1c).

To assess the amount of HA secreted by these cell lines, a carbazole assay was performed to quantify the HA purified from the conditioned media. All five subterranean species secreted higher amounts of HA than their aboveground controls (Fig. 1d). Considering that carbazole assay may also react with other sugars, we used HA ELISA method to quantify the HA. The media conditioned by BMR cells contained more HA than the media conditioned by mouse cells (Supplementary Fig. S1a). Additionally, we quantified the intensity of HA on a gel, and the results showed a similar trend (Supplementary Fig. S1b), with all three assays supporting the conclusion that subterranean species produce higher amounts of HA.

We then compared the expression levels of three major factors responsible for the tissue levels of HMM-HA: the hyaluronan synthase 2 (*HAS2*), hyaluronidase-1 (*HYAL1*), and hyaluronidase −2 (*HYAL2*) genes. HAS2 is responsible for the synthesis of HMM-HA. HYAL2 hydrolyzes

HMM-HA into intermediate length HA[1,23], and HYAL1 further degrades small molecules of HA into tetrasaccharides[24]. The BMR cells expressed significantly higher levels of HAS2 mRNA and lower HYAL2 than mouse cells (Fig. 1e, f), both contributing to the accumulation of HMM-HA. NMR and DMR cells expressed higher levels of both HAS2 and HYAL2, but the fold-change of HAS2 was much higher than that of HYAL2 (Fig. 1e, f). Meanwhile, both NMR and DMR cells expressed dramatically lower levels of HYAL1 (Fig. 1g). Interestingly, EM and SNM had lower expression of both HYAL1 and HYAL2 compared with the shrew (Fig. 1f, g). These results suggest that subterranean species secrete higher molecular mass and higher amount of HA by differentially regulating the expression of *HAS2*, *HYAL2* and *HYAL1* genes.

### Subterranean species have higher HA abundance in tissues due to differential expression of HA synthases and hyaluronidases

To evaluate the amount of HA generated in vivo, we compared the levels of HA in different tissues of subterranean and aboveground species. Skin had the most abundant HA. In humans, skin contains 50% of total HA in the body[25]. Furthermore, heart and kidney are the two organs that have been shown to contain high amounts of HA in rodent species[1]. Cryosections of skin, heart, and kidney were stained with fluorescently labeled hyaluronan binding protein (HABP), and quantified. All six subterranean species had significantly higher amounts of HA in the skin than their aboveground controls (Fig. 2a). With a few exceptions, similar results were observed in heart and kidney (Supplementary Fig. S2a and S3a). All samples showed negative staining after HAase treatment, demonstrating the specificity of HA detection (Supplementary Fig. S4). These results suggest that subterranean species produce higher amounts of HA in their tissues.

We next examined the expression levels of hyaluronan synthase HAS2, hyaluronidase HYAL2, and HYAL1 by RT-qPCR. BMR and TMV had lower expression of both HYAL1 and HYAL2 (Fig. 2b and Supplementary Fig. S5a); NMR and DMR had dramatically higher HAS2 expression. Surprisingly, although both EM and SNM had higher expression of HAS2, expression levels of both HYAL2 and HYAL1 were also higher than in the shrew (Fig. 2b and Supplementary Fig. S5a). It is possible that additional regulatory factors contribute to the higher abundance of HA in the skin of EM and SNM. These results suggest that subterranean species differentially regulate the expression of HAS2 and HYAL2 in their skin to generate more abundant HA.

Interestingly, we found that the expression of HAS2, HYAL1 and HYAL2 was regulated differently in the heart and kidney than in the skin. For example, in the heart, BMR had lower expression of all three enzymes, and NMR had significantly lower expressed HYAL1 (Supplementary Fig. S2b). In the kidney, TMV had a clear trend of higher expression of HAS2, although it was not significant due to large variation (Supplementary Fig. S3b). NMR had a higher expression of HYAL2 and a drastically lower expression of HYAL1 in the kidney (Supplementary Fig. S3b). These results suggest that even though HA accumulates in different tissues of subterranean species, their regulatory mechanisms are tissue specific.

### Evolution of HA degrading genes in subterranean species

As we demonstrated above, the abundance of HA in the subterranean species differs across tissues. Therefore, to systematically assess the regulation of HMM-HA in different tissues, we performed RNA sequencing (RNAseq) on samples from 6 tissues (skin, brain, lung, heart, kidney and liver) across 10 species (6 subterranean and 4 aboveground species): BMR, NMR, DMR, TMV, EM, SNM, rat, mouse, GP, and shrew (Fig. 1a). Hierarchical clustering of overall expression similarity matrices derived from 8417 orthologs across all species showed that samples are clustered mainly by tissue rather than by species (Supplementary Fig. S6a). Therefore, the interspecies gene expression comparisons were performed within each tissue. The expression data of samples from the same tissue were grouped and

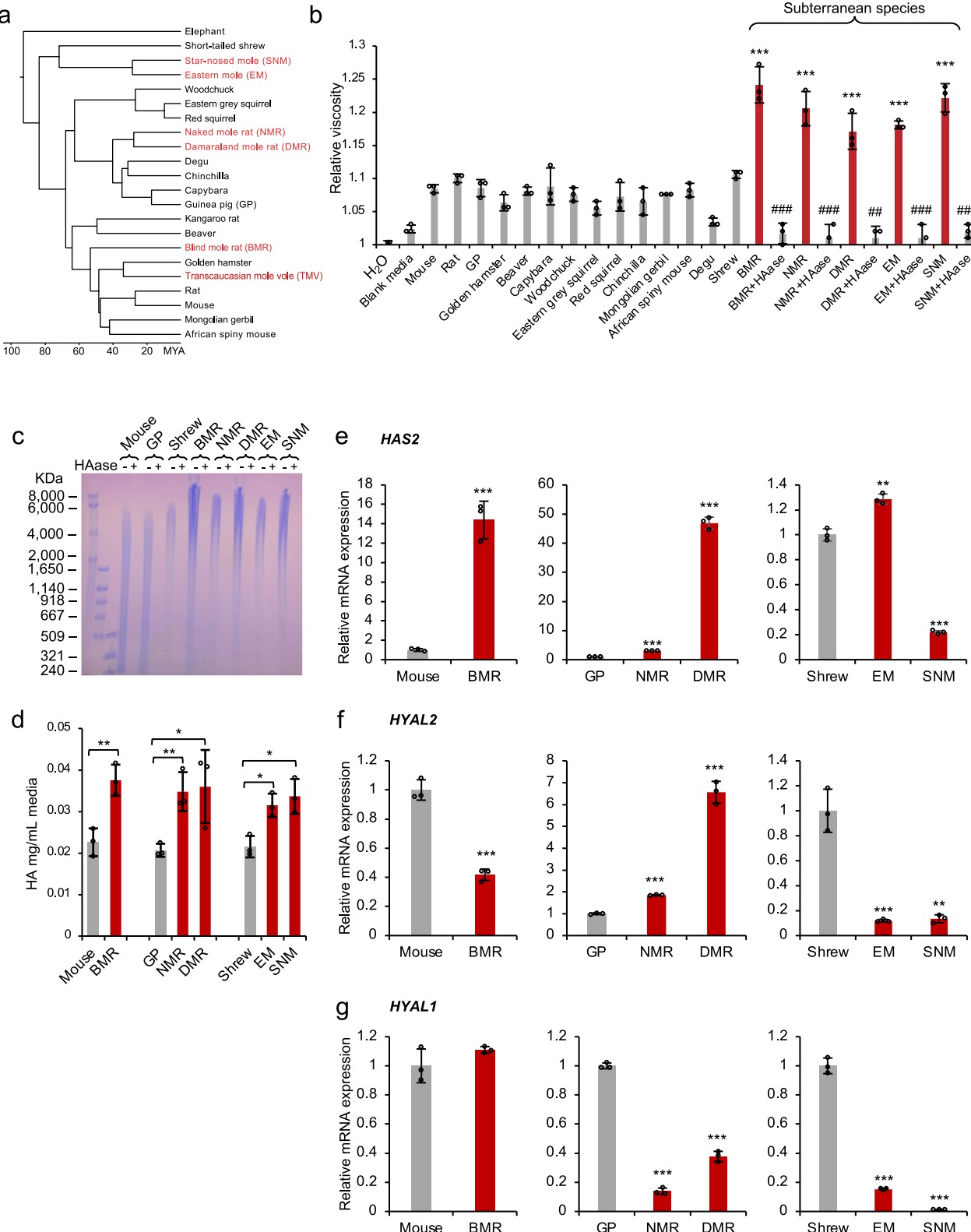

filtered (removing genes with unavailable expression data or no count in more than two thirds of samples) and normalized separately. As a result, we obtained 16,204, 14,832, 16,629, 15,725, 16,502 and 16,136 genes in skin, brain, lung, heart, kidney and liver, respectively. The normalized expression data showed similar median expression value for each sample (Supplementary Fig. S6b).

We first examined the expression of HAS2, HYAL1 and HYAL2 from the RNAseq data, which was generally consistent with the qPCR results (Supplementary Fig. S5b). Particularly, EM and SNM HAS2 were more highly expressed in the skin, heart, and kidney compared to the shrew as demonstrated in both qPCR and RNAseq data (Fig. 2b, Supplementary Fig. S2b, S3b, and S5b). The expression levels of HAS2 were

**Fig. 1 | Fibroblasts of subterranean species secret abundant HMM-HA.**
**a** Phylogenetic tree of subterranean species used in this study. Subterranean species are shown in red. **b** Fibroblasts of subterranean species produce viscous culture media. The viscosity of water, blank media, or media conditioned with skin fibroblasts of different species was measured. Conditioned media was harvested after 10 days. For cells of subterranean species (BMR, NMR, DMR, EM, and SNM), conditioned media was treated with hyaluronidase (HAase) to show that the viscosity is due to the presence of HA. Experiment was repeated three times. ## $P < 0.01$, ### $P < 0.001$ compared between HAase treated and untreated media, ***$P < 0.001$ media conditioned with cells of subterranean species compared to blank media, unpaired two-sided $t$ test. **c** Pulsed-field gel electrophoresis of HA purified from conditioned media of different species. Cells were kept confluent for 10 days and HA was purified from equal volumes of conditioned media. Each sample was either run directly or pre-treated with HAase overnight. The experiment was performed three times with similar results. **d** Levels of HA secreted by cells of different species measured by carbazole assay. The assay was performed on equal volumes of conditioned media. Reverse transcription quantitative PCR (RT-qPCR) showing expression levels of *HAS2* (**e**), *HYAL2* (**f**) and *HYAL1* (**g**) genes in skin fibroblasts of different species. GP guinea pig, BMR blind mole rat, NMR naked mole rat, DMR Damaraland mole rat, EM eastern mole, SNM star-nosed mole. (**b**, **d**, **e**, **f**, and **g**) experiments were repeated three times. Data are mean ± SD. *$P < 0.05$, **$P < 0.01$, ***$P < 0.001$ unpaired two-sided $t$ test. Source data are provided as a Source Data file.

higher in the skin of NMR and DMR than in GP (Fig. 2b, Supplementary S5b). The expression levels of HYAL1 were significantly lower in the heart and kidney of NMR than in GP (Supplementary Fig. S2b, S3b, and S5b). Furthermore, HA is synthesized by three HA synthases (HAS1-3) and degraded by six hyaluronidases (HYAL1-6) and two newly discovered HA-degrading enzymes CEMIP (KIAA1199)[26] and CEMIP2 (TMEM2)[27,28]. The amount and size of tissue hyaluronan are determined not only by the activities of synthases and hyaluronidase, but also by the metabolic status of tissue UDP-sugars, especially UDP-GlcNAc and UDP-GlcUA[29] (Fig. 3a). Previous study also identified *HYAL3* and *TNFAIP6* as being more highly expressed in NMR[22]. Therefore, we investigated the expression changes of genes involved in UDP-GlcNAc synthesis, UDP-GlcUA synthesis, hyaluronan synthesis, hyaluronan degradation, hyaluronon binding, and TNFAIPs. For each tissue, we grouped all the subterranean species into one group and all the aboveground species into another in order to identify genes with convergent gene expression changes. Interestingly, we found a large number of HA binding genes to be more highly expressed in subterranean species, but *HABP2* gene, which we used to stain for HA in immunofluorescence, did not show significant differences across species (Fig. 3b). HA degrading genes including HYAL3, CEMIP, and CEMIP2 had lower expression in multiple tissues of subterranean species (Fig. 3b). These results suggest that impaired HA degradation may have contributed to the accumulation of high amount and molecular mass of HA in subterranean species.

We next examined the evolution of genes involved in the degradation of HA. By using branch-site models (model A)[30], we identified five sites in HYAL2 as positively selected sites in the ancestor of SNM and EM ($2\Delta l = 15.0219$, df = 1, $P = 0.0001$), including two sites (284 K 0.989 and 348 S 0.932) with Bayes empirical Bayes (BEB) values > 0.9 and three sites (106H 0.689, 280 N 0.501, and 352 A 0.604) with BEB values > 0.5 (Fig. 3c). All the positively selected sites are located in the GHF domain, which is responsible for the hydrolase activity and HA degradation. Interestingly, an amino acid replacement A284G occurred in the ancestor of NMR and DMR that overlapped with the strongest positive selection sites A284K identified in SNM and EM (Fig. 3c). Two parallel amino acids replacements, Q149E (shared by NMR and EM) and R194H (shared by BMR and DMR), were found in the GHF domain of HYAL1 (Fig. 3d).

Gene loss has been reported in BMR and NMR contributing to adaptation to subterranean environments[31], we therefore further examined pseudogenization of HA related genes in subterranean species. Consistent with previous study, only HYAL3 was found pseudogenized (frameshifts and premature stop-codons) in BMR and showed a significant relaxed selection constraint (RELAX, $k = 0.34$, $P = 0.02$). Interestingly, we found that two subterranean species close to BMR, the plateau zokor (*Myospalax baileyi*) and hoary bamboo rat (*Rhizomys pruinosus*), have also lost HYAL3 due to either shared or independent mutations (Fig. 3e). Collectively, these results suggest that hyaluronidase genes are frequently under selection in subterranean species.

## Differential expression and sequence changes in HA synthases and hyaluronidases contribute to HMM-HA in subterranean species

To mechanistically assess the contribution of differential expression and sequence changes in HA related enzymes to accumulation of HMM-HA in subterranean species, we overexpressed HAS2, HYAL1, or HYAL2 from different species in 293 T or HeLa cells and tested their effect on HA synthesis and degradation. Based on our analysis, BMR fibroblasts had a 14-fold higher endogenous expression level of HAS2 than mouse cells (Fig. 1e), therefore, we transfected mouse or BMR HAS2-encoding plasmids at a 1:14 ratio to mimic the endogenous expression difference. To control for transfection efficiency, RT-qPCR was performed to detect the expression of transfected genes, which showed significantly higher levels of BMR HAS2 than mouse HAS2 as expected, and similar expression levels of all the other transfected genes (Supplementary Fig. S7). Two days after transfection, HA was purified from the conditioned media and subjected to electrophoresis. The 293 T cells expressing 14-fold higher levels of BMR HAS2 yielded more abundant HA (Supplementary Fig. S8a). In HeLa cells, the cells expressing high level of BMR HAS2 generated not only significantly more abundant, but also larger-sized HA than cells expressing low level of mouse HAS2 (Fig. 4a). Strikingly, even when equal amount of mouse and BMR HAS2 were transfected, the size of HA secreted by BMR HAS2-expressing cells was still larger than that secreted by mouse HAS2-expressing cells (Fig. 4a, lane 4 vs. lane 5). Interestingly, the HAS2 gene is highly conserved between mouse and BMR, with only four amino acid changes (Supplementary Fig. S9a). Therefore, this result suggests that the four mutations in BMR HAS2, combined with its higher expression, contribute to the accumulation of HMM-HA in BMR. Similarly, cells expressing SNM HAS2 also generated larger-sized HA than the cells expressing shrew HAS2 (Fig. 4b). These results suggest that the sequence changes and differential expression levels of HAS2 contribute to accumulation of HMM-HA in subterranean species.

We next compared the HA degradation between subterranean and aboveground species. Commercial HMM-HA was incubated with confluent fibroblasts of five subterranean species or three aboveground control species for two days. After incubation, more HA remained after incubation with the cells of subterranean species than aboveground species (Fig. 4c). The result remained the same when the HA was incubated with growing cells (Supplementary Fig. S8b). These results suggest that cells of subterranean species have degraded HA less.

We then examined the activities of HA-degrading enzymes, HYAL2 and HYAL1, from different species. We overexpressed either HYAL2 or HYAL1 from mouse, BMR, shrew, or SNM in 293 T cells. Commercial HA was incubated with the transfected cells for two days, extracted and subjected to electrophoresis. The shrew HYAL2 induced more degradation of commercial HA than the SNM HYAL2 (Supplementary Fig. S8c, d), suggesting that the HYAL2 in subterranean species has weaker HA-degrading activity. We performed the same experiment in HeLa cells. While no differences were observed after two days of

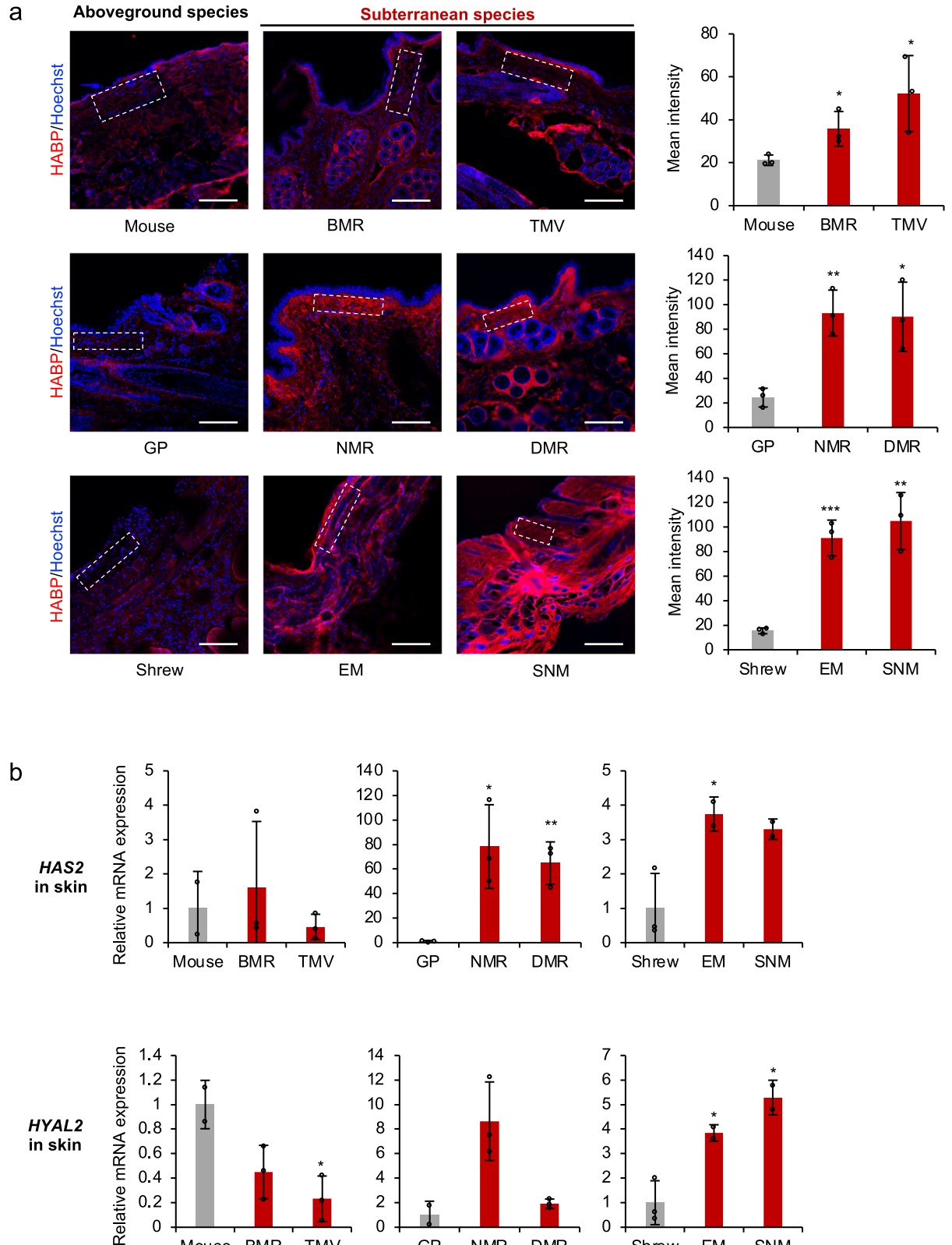

incubation, when the incubation was extended to three days, significant amount of commercial HA was degraded. Strikingly, mouse and shrew HYAL2 degraded more HA, resulting in smaller size of remaining HA compared to BMR and SNM, respectively (Fig. 4d). As expected, cells expressing HYAL1 of different species did not show significant difference in HA degradation (Fig. 4d and Supplementary Fig. S8e), probably because HYAL1 is an intracellular enzyme[32]. Taken together, these results suggest that HYAL2 has lower activity in subterranean species contributing to accumulation of HMM-HA.

**Structural influences of EM and SNM HYAL2 mutations**

To analyze the structural consequences of the five residues identified in the EM and SNM HYAL2, SWISS-MODEL structures were generated. These structures were compared to human HYAL2 predicted by

**Fig. 2 | Skin of subterranean species contains higher levels of HA. a** Cryosections of skin of different species were stained with hyaluronan binding protein (HABP, red) and Hoechst (blue). Representative immunofluorescence images are shown at 20 × magnification (left). Scale bar, 50 μm. Quantification of HABP fluorescence intensity (right panel) was performed on the similar regions of each sample as indicated by the squares with dashed outline (left panel). For each sample, cumulative intensities were normalized to the area of selected regions to control the volume. Data are mean ± SD of three independent areas. *$P < 0.05$, **$P < 0.01$, and

***$P < 0.001$ comparing subterranean species with their aboveground controls by unpaired two-sided $t$ test. **b** RT-qPCR showing expression levels of *HAS2* and *HYAL2* genes in the skin comparing subterranean species with their aboveground controls. Experiments were performed using three biological replicates except mouse, EM and SNM, which had two biological replicates. Red bars indicate subterranean species. Data are mean ± SD. *$P < 0.05$, **$P < 0.01$ by unpaired two-sided $t$ test. Source data are provided as a Source Data file.

AlphaFold[33]. The overall structure of AlphaFold human HYAL2 and SWISS-MODEL EM and SNM HYAL2 with highlighted residues from human and two moles is shown; all five residues are at the exterior and located at three different positions (Fig. 5a).

Human residues Thr348 and Val352 and mole Ser351 and Ala355 are found at the kink of an alpha-helix and have interactions with a neighboring loop. The chemically similar mutations are far from the active site and are not likely to directly influence the overall structure or enzymatic activity (Fig. 5b, c). However, these residues reside in the EGF-like domain and may play a role in protein-protein interactions and regulatory processes of HYAL2[34].

Human residue Ala284 is located at the C-terminal end of an alpha-helix at the exterior of HYAL2. Human Ala284 is chemically very different from Lys287 in the moles. This mutation is unlikely to influence the structure of HYAL2 due to its exterior positioning, but may alter the post-translational modification state, membrane localization, or interaction partners of mole HYAL2. Human residue Arg280 is located within an alpha-helix and the side chain extends into a hydrophobic pocket lined by Leu23, Leu276, and Val377. Human Arg280 extends far into the hydrophobic pocket and the charged guanidinium group has unfavorable interactions in the nonpolar region. However, mole Asn283 in the moles rotates outside of the hydrophobic pocket. Therefore, the mole residue Asn283 removes an unfavorable interaction seen in human Arg280 (Fig. 5d, e). Human Leu106 and mole His109 are found at the N-terminal end of an alpha-helix within the interface between two alpha-helices near the active site. In humans, the strongest interactions include Val104, His109 and Phe186. In the structure model, mole His109 rotates to the exterior but may be capable of rotating inwards to engage in a T-shaped Pi-stacking interaction with Phe189 located approximately 3.6 Å away on the adjacent alpha-helix and form a pi-stacking cascade along with His112 (Fig. 5f, g). This may stabilize the helix-helix interaction affecting the nearby catalytic residues (Asp136 and Glu138) in EM and SNM.

HYAL2 is a highly post-translationally modified enzyme. The first 20 amino acids of HYAL2 are predicted to be a signal sequence that directs the protein to the endoplasmic reticulum[35]. In the endoplasmic reticulum, HYAL2 is glycosylated and trimmed to the mature protein (residues 21–448) with the addition of a glycophosphatidylinositol (GPI) anchor onto Gly448[35]. Trimming removes two large alpha-helices (residues 1–20 and 449–473) that extend out from HYAL2. Removing these two alpha-helices will expose mole Lys287 to the solvent more and may facilitate its membrane localization, interaction partners, or post-translational modification state (Fig. 5a, e). This residue is also located near the GPI anchor and is oriented towards the membrane (Fig. 5e). Therefore, residue Lys287 in the moles may influence its interaction partners or localization at the membrane when compared to human Ala284. Glycosylation is important for hyaluronidases' enzymatic activity. Enzymatic removal of 50% of glycosylation from human HYAL1 reduced activity by 40%[36]. HYAL2 is predicted to undergo glycosylation at three asparagine residues. In humans, Asn103 has been confirmed to be glycosylated and Asn74 and Asn357 are predicted to be glycosylated[35]. The glycosylation sites Asn103 and Asn357 are near the mutant residues 109 and 351 + 355 in the moles, respectively (Fig. 5c, g). Asn103 is located adjacent to residue 106.

Therefore, the mutation at residue 106 may influence the local structure and the glycosylation state of Asn103. HYAL2 residue 357 glycosylation is located at an interface between the catalytic domain and the HyalEGF-like module. Although residues 348 and 353 have chemically similar residues between human and moles, slight structural changes in the HyalEGF-like module may influence the degree of Asn357 glycosylation. Collectively, the mutations observed in EM and SNM may potentially influence the protein stability and post-translational modifications of HYAL2.

To experimentally test if the amino acid substitutions in HYAL2 affect its enzymatic activity, we generated five plasmids corresponding to each of the five mutation sites in EM and SNM HYAL2. By incubating commercial HA with 293 T cells expressing HYAL2 plasmids, we found that, when the SNM HYLA2 was mutated to the amino acid sequence of shrew in each site, the HA-degrading ability became stronger (Supplementary Fig. S8f, g). In HeLa cells, when incubated for three days, mutants H106L, K284A, and A352V resulted in smaller sized HA compared with wild-type HYAL2 (Fig. 5h). These results suggest that at least the three unique mutations of EM and SNM HYAL2 resulted in a weaker HYAL2 activity, which is likely to contribute to the accumulation of HMM-HA. A positively selected site G284 was also identified in NMR and DMR HYAL2 (Fig. 3c). We tested the effect of this mutation on NMR HYAL2. We did not observe difference when HA was incubated for two days, but the size and amount of HA was smaller after three days of incubation (Fig. 5i), suggesting that the substitution in G284 also contributes to the weaker HYAL2 in NMRs.

## HMM-HA coevolves with ECM in subterranean species

HA is a major component of ECM and it interacts with other ECM components such as collagens, therefore the convergent abundant HMM-HA phenotype may have coevolved with other changes in the ECM. To systematically examine the evolutionary landscape of ECM and its relationship with HA, we compiled a gene set involved in ECM function (2932 ECM-related genes in total) (Fig. 6a) and examined the genes under positive selection and/or carrying convergent/parallel amino acid substitutions. In addition to subterranean species in this study, we included another two species, plateau zokor and hoary bamboo rat, both of which have high quality genomes[37,38]. We performed two independent evolutionary analyses with or without these two species, which generated two datasets with (1, Fig. 6b) or without plateau zokor and bamboo rat (2, Supplementary Fig. S10). In both datasets, we detected fewer positively selected genes (PSGs, FDR < 0.1) and convergently evolving genes (CEGs, shared by >3 species) in subterranean species (A, 430 PSGs and 697 CEGs; B:188 PSGs and 101 CEGs) compared to aboveground lineages (A:320 PSGs and 1395 CEGs; B:323 PSGs and 347 CEGs; Supplementary Data 1 and 2). This pattern is consistently observed across each subterranean-aboveground species pair, except for bamboo rat, which showed more PSGs than the aboveground species (Fig. 6b). Despite this, all subterranean species showed lower numbers of CEGs, lower ratio of convergent-to-divergent sites (C/D ratio)[39], and lower numbers of genes with both positively selected and convergent sites (Fig. 6b). This result is consistent with previous reports that species living under harsh environments generally have fewer PSGs due to a smaller effective population size[40–44].

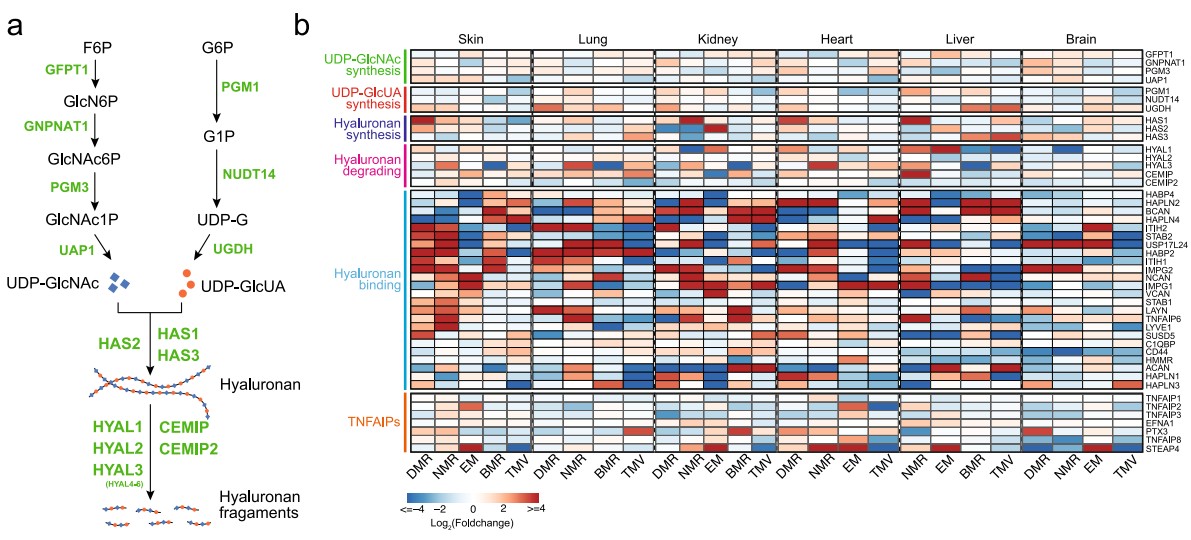

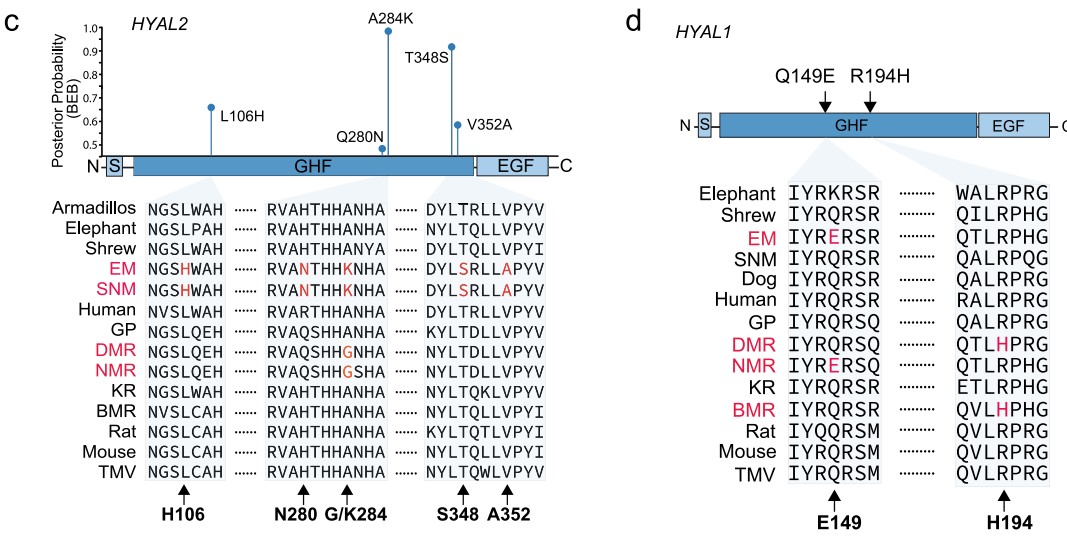

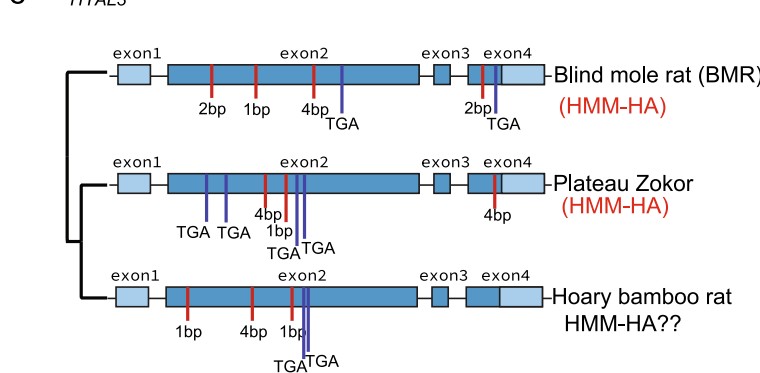

We found changes in three genes associated with basement membrane (BM) that are of particular interest. NID1 (Nidogen 1) is a BM glycoprotein that interacts with type IV collagen and lamina network[45]. Laminin-332, a type of laminin network, is composed of laminin subunits α3 and β3, encoded by LAMA3 and LAMB3, respectively. Two amino acid substitutions were observed in NID1 of subterranean species. The N-terminal Isoleucine-to-Valine is shared by NMR, BMR, plateau zokor, SNM, and TMV, and the C-terminal Aspartate-to-Glutamate is shared by DMR, SNM, and TMV (Supplementary Fig. S9b). In addition, NID1 was also detected as a PSG in DMR ($P = 0.0026$, FDR = 0.0861; Supplementary Data 2). LAMA3 and LAMB3 were also under positive selection in several lineages including EM (Supplementary Data 2: LAMA3: $P = 3.79e-11$, FDR = 1.43e-08; LAMB3: $P = 7.24e-07$, FDR = 0.0002), DMR

**Fig. 3 | Comparative analysis of gene expression and molecular evolution of genes involved in HA synthesis and degradation in subterranean species. a** HA synthesis and degradation pathway. Genes responsible for each enzymatic reaction are labeled in green. F6P Fructose 6-phosphate, GlcN6P Glucosamine-6-phosphate, GlcNAc6P N-acetyl-glucosamine 6-phosphate, GlcNAc1P N-acetyl-glucosamine 1-phosphate, UDP-GlcNAc Uridine diphosphate N-acetylglucosamine, G6P Glucose 6-phosphate, G1P Glucose 1-phosphate, UDP-G Uridine diphosphate glucose, UDP-GlcUA Uridine diphosphate glucuronic acid. **b** Heatmap showing the expression fold changes for HA-related genes. Genes are grouped into six sets corresponding to their function, including HA substrate metabolic pathways (UDP-GlcNAc synthesis and UDP-GlcUA synthesis), HA synthesis, HA degradation, HA binding, and TNFAIP6 related genes. The fold change values were calculated by comparing each

subterranean species to their closest aboveground species. For tissues where the closest aboveground species samples are not available, rat samples were used. **c** Diagram on top shows the domain structure of HYAL2 and corresponding Bayes Empirical Bayes posterior probabilities (PP) of positive selection in the common ancestor of SNM and EM, with sites predicted to be under positive selection in blue (PP > 0.5). Amino acid alignment of the positively selected sites is shown below. S: Signal peptide. GHF Glycoside hydrolase family 56 (Pfam). EGF EGF-like domain. **d** Convergent amino acid replacements between BMR and DMR or between NMR and EM. **e** Diagram showing the disruptive mutations identified in GHF domain of HYAL3 of BMR and two closely related species, Bamboo rat and Plateau zokor. Red bars: indel; Blue bars: stop-codon.

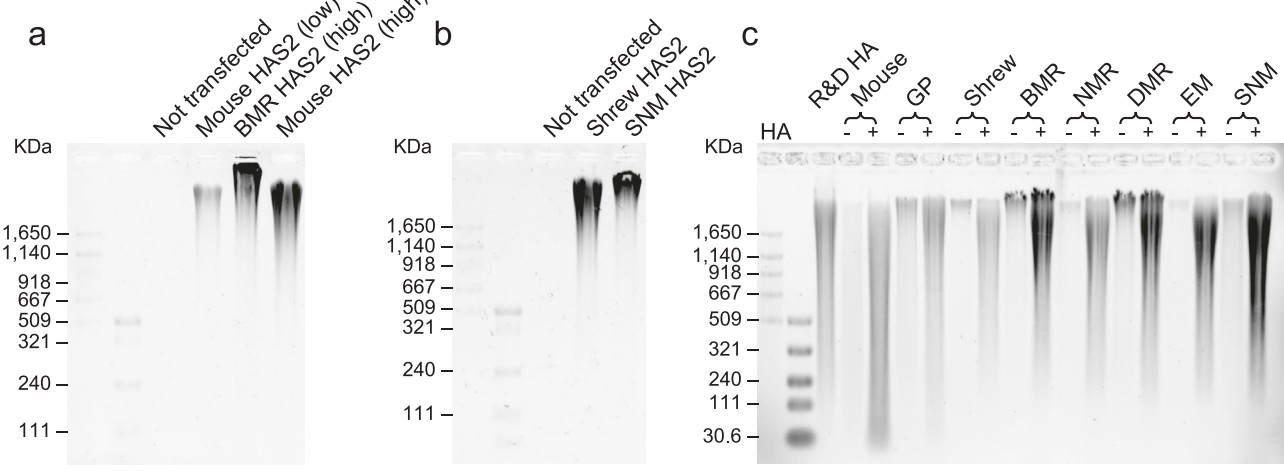

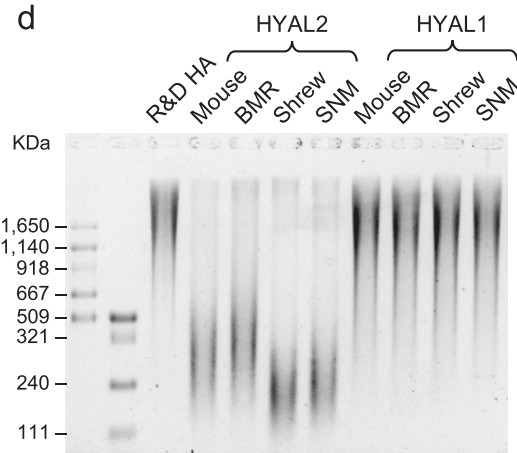

**Fig. 4 | Effects of HAS2 and HYAL2 changes on HA synthesis and degradation. a** HeLa cells were transfected with low amount of mouse HAS2, high amount of BMR HAS2 (14-fold of mouse HAS2), or high amount (equal to BMR HAS2) mouse HAS2 plasmids. Two days after transfection, HA was purified from the media and analyzed by gel electrophoresis. **b** HeLa cells were transfected with equal amounts of shrew or SNM HAS2 plasmids and incubated for two days. HA was purified and run on an agarose gel. **c** Media containing 20 μg/mL commercial HMM-HA was

incubated with confluent fibroblasts from each species. Two days after incubation, HA was purified from the media and run on an agarose gel. For each cell line, a sample without commercial HA was included to control the endogenous HA. **d** HeLa cells were transfected with either HYAL2 or HYAL1 of mouse, BMR, shrew, or SNM. Transfected cells were cultured in media containing 10 μg/mL commercial HA. Three days after incubation, HA was purified and run on an agarose gel. All experiments were performed three times with similar results.

(Supplementary data 1: LAMA3: $P = 4.07e\text{-}11$, FDR = $2.23e\text{-}08$. Supplementary data 2: LAMA3: $P = 2.67e\text{-}10$, FDR = $9.81e\text{-}08$; LAMB3: $P = 6.10e\text{-}06$, FDR = 0.0006), the ancestor of NMR/DMR (Supplementary Data 2: LAMA3: $P = 1.79e\text{-}09$, FDR = $7.04e\text{-}07$; LAMB3: $P = 0.0010$, FDR = 0.0630), and plateau zokor (Supplementary Data 1 and 2: LAMA3: $P = 4.10e\text{-}08$, FDR = $2.41e\text{-}06$). Notably, LAMB3

was still detected as a PSG when all the subterranean species were combined ($P = 0.0001$, FDR = 0.06; Supplementary Data 2). As laminin-332 is predominantly located in the skin, these results suggest that the adaptive evolution of these three genes may have contributed to the special BM structure in the skin of the subterranean species.

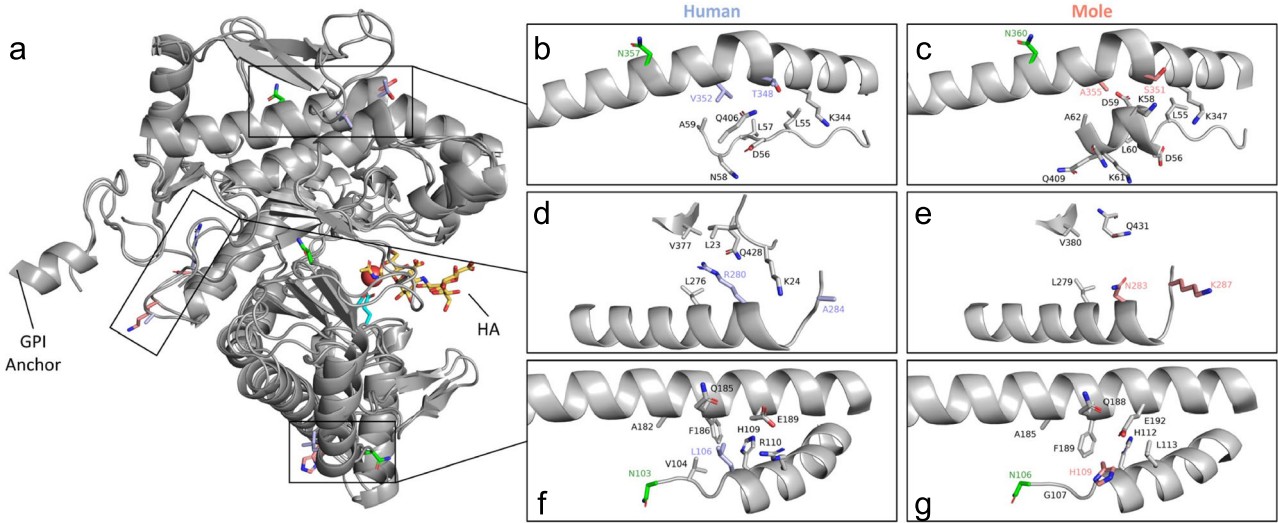

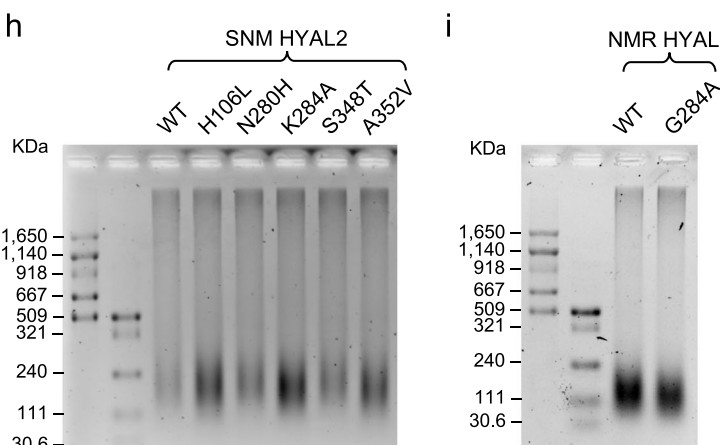

**Fig. 5 | Structural and functional analysis of EM and SNM HYAL2 mutations.**
**a** Cartoon representation of human HYAL2 ranging from residues 21 to 448 as predicted by AlphaFold. For the positively selected sites identified in moles, the corresponding human residues are shown in orange and mole residues are shown in purple. Catalytic residues are shown in cyan and glycosylated residues in green. A molecule of HA tetramer (yellow) from bee venom HYAL (PDB code: 1FCV) was mapped to human HYAL2 using the PyMOL align function. **b**, **c** Zoomed in image of mutant residues 348 and 352 comparing human and moles. **d**, **e** Zoomed in image of residues 280 and 284 comparing human and moles. **f**, **g** Zoomed in image of residue 106 comparing human and moles. Only neighboring residues to the amino acids under positive selection in moles are shown. **h**, **i** Effect of SNM and NMR HYAL2 mutations on HA degradation. **h** HeLa cells were transfected with either wild-type (WT) SNM HYAL2 plasmid, or each of its five mutants. **i** HeLa cells were transfected with either NMR HYAL2 WT or the G284A mutant. Cells were incubated with media containing 10 µg/mL commercial HA for three days, followed by HA extraction an electrophoresis. **h**, **i** experiments were performed three times with similar results.

PSGs tend to have greater tissue bias than non-PSGs[46], we therefore examined whether the PSGs identified in the ECM of subterranean species display tissue bias. The tissue bias, measured by the statistic $\tau$[47], was calculated for each species using transcripts per million (TPM). As expected, PSGs had higher tissue bias when compared to non-PSGs (Fig. 6c). Interestingly, PSGs identified in each subterranean species had greater tissue bias compared to their aboveground controls (Fig. 6c). This is particularly the case for the genes with $\tau > 0.25$, which are usually considered to be tissue-specific genes. These results suggest that the selection on the ECM genes of subterranean species may drive the adaptations of specific tissues to the subterranean environment including the accumulation of HMM-HA.

We next examined the functional enrichment of differentially expressed genes (DEGs). All genes that are consistently higher or lower expressed in subterranean species for each tissue were analyzed for functional enrichment. Strikingly, the most significantly enriched GO categories for lower expressed genes in lung were ECM related (e.g.,

GO:0030198 extracellular matrix organization, FDR = 5.28e-14) (Supplementary Data 3). To characterize the regulation of ECM genes in subterranean species, we compared the numbers of enriched ECM-related GO terms ($P < 0.05$) in each tissue. As a result, skin (for higher expressed genes) and lung (for lower expressed genes) had the greatest number of ECM-related GO terms, which showed opposite regulatory directions (Fig. 6d). To rule out the possible inaccuracy due to over-representation analysis, we employed GSEA and Generally Applicable Gene-set Enrichment (GAGE)[48] analysis to validate this pattern, and the results displayed similar patterns (Fig. 6e). While only a limited number of ECM-related GO terms were enriched in skin, all other tissues had much more GO terms showing enrichment for lower expressed genes. This pattern is exemplified by BM associated collagen genes, especially COL4A1 and COL4A2. We found that COL4A1 and COL4A2 had consistently lower expression in the subterranean lung but not in the skin samples, compared to aboveground samples (Supplementary Fig. S11). Other tissues such as brain, heart, and kidney

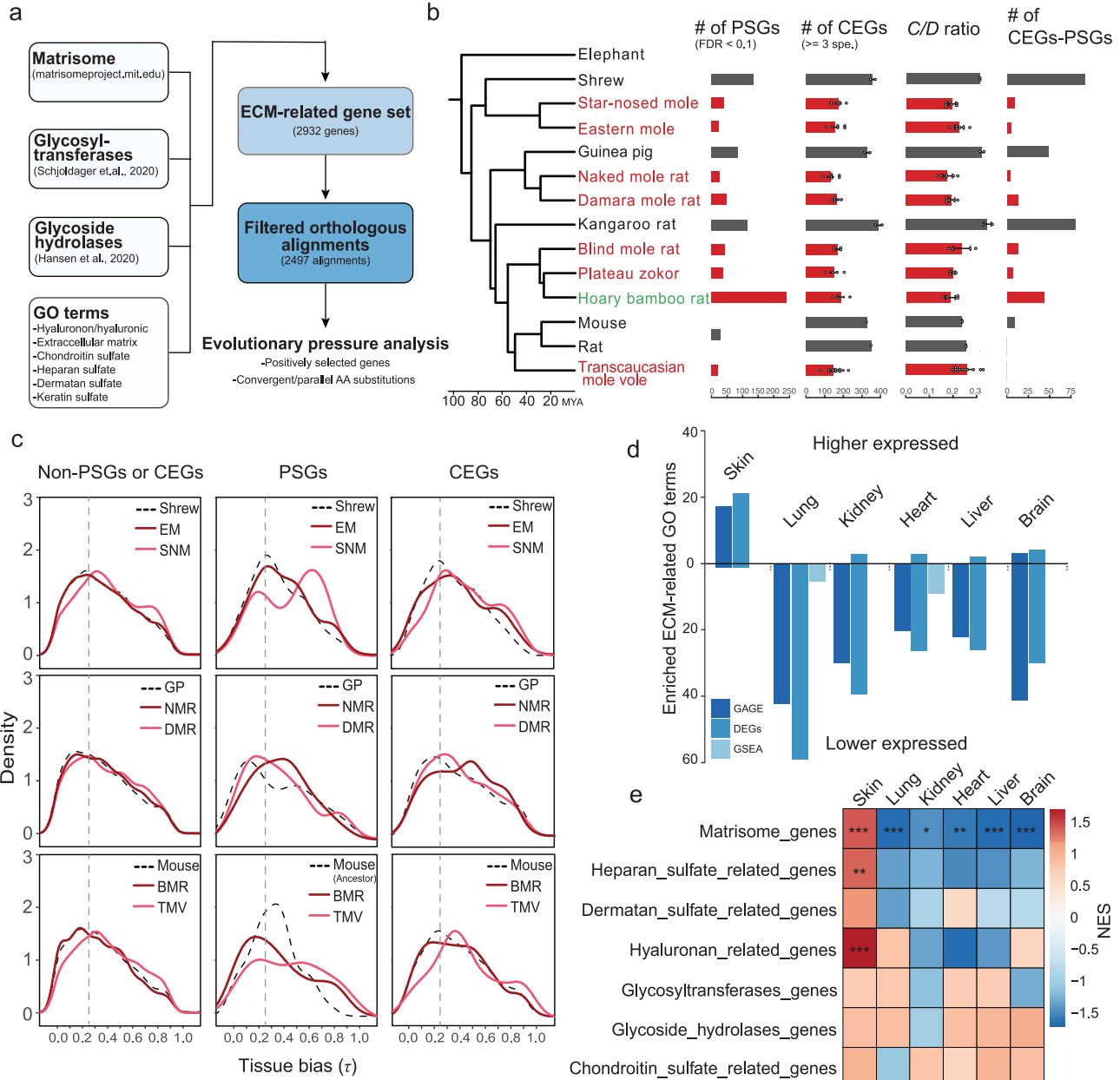

**Fig. 6 | Selection pressure analysis and regulatory changes in ECM-related genes in subterranean species. a** Diagram showing the analysis workflow. **b** Phylogenetic tree of species in this study and the numbers of positively selected genes (PSGs), convergently evolving genes (CEGs), convergent-to-divergent site ratio (*C/D* ratio; >3 species) and genes with both positive selection sites and convergent sites (CEGs-PSGs). Data are mean ± SD of CEGs and *C/D* ratio resulting from different four species combinations. The number of four-species combinations are: shrew *n* = 2, star-nosed mole *n* = 6, eastern mole *n* = 6, guinea pig *n* = 2, naked mole rat *n* = 6, Damaraland mole rat *n* = 6, kangaroo rat *n* = 2, blind mole rat *n* = 4, plateau zokor *n* = 4, hoary bamboo rat *n* = 4, mouse *n* = 1, rat *n* = 1, and Transcaucasian mole vole *n* = 12. Subterranean species are shown in red. **c** Tissue bias (measured by the statistic τ) distribution of PSGs, CEGs and non-PSGs or -CEGs in subterranean species (light and dark red lines) and their corresponding aboveground control species (dashed black line). PSGs and CEGs are combined gene sets from analysis

with and without plateau zokor and hoary bamboo rat. Vertical dashed lines indicate τ = 0.25. Genes with τ > 0.25 were considered tissue-specific genes. **d** The number of ECM-related GO terms enriched by genes that are consistently up- or down-regulated across subterranean species compared to aboveground controls. Different methods show a similar pattern. GAGE: Generally Applicable Gene-set Enrichment analysis using the normalized expression matrix as input; DEGs: over-representation analysis using differentially expressed genes identified by comparing all subterranean species to all above ground species; GSEA: gene set enrichment analysis using the ranked fold changes from DEG analysis. **e** Heatmap showing NES scores from gene set enrichment analysis for the custom ECM-related gene sets. The *P* values are calculated using a one-sided permutation-based approach and were adjusted for multiple testing with Benjamini-Hochberg (BH) correction. *\**P.adjust* < 0.05, *\*\*P.adjust* < 0.01, *\*\*\*P.adjust* < 0.001.

also showed similar trend with some outlier samples. These results are consistent with our evolutionary analysis results that BM related genes had been positively selected across various subterranean species. We further validated this pattern by performing GSEA analysis to our custom ECM gene sets. Consistent with our GO term analysis, the

subterranean skin samples tended to show higher expression of ECM genes, especially "Matrisome genes", while other tissues tended to show lower expression. Taken together, these results suggest that skin and lung are two major tissues that have experienced ECM remodeling during adaptation to subterranean environment. Further studies are

needed to explore the relationship between ECM, BM remodeling and HMM-HA accumulation.

## Discussion

In the present study, we revealed that several unrelated subterranean species, but not their aboveground relatives, evolved to produce HMM-HA. Furthermore, the skin, heart, and kidney tissues of subterranean species secrete higher amount of HA compared to related aboveground species. This is achieved by differential expression of HA synthases and hyaluronidases combined with convergent or positively selected mutations on hyaluronidase genes. Therefore, abundant HMM-HA is a common phenotype for subterranean species, suggesting a broad adaptive benefit of HMM-HA for subterranean lifestyle.

An intriguing characteristic of HA metabolism is its fast turnover. One third of HA in the body undergoes turnover daily[49]. The size of HA is determined by the rate of synthesis and degradation, which are carried out by several hyaluronidases including HYAL1, HYAL2, and PH20[50,51]. Our study demonstrated that, although HMM-HA is shared by all subterranean species investigated, each species achieves the HMM-HA through a different mechanism (Fig. 7). It was proposed that two unique mutations of NMR HAS2 contribute to the HMM-HA[1]. A comparative study later revealed that one of these two mutations is only shared by a few species of African mole-rats but not by other subterranean species[21]. We systematically analyzed mutations in HA synthesis and degrading enzymes and revealed that positively selected or convergent mutations of HYAL1 and HYAL2, shared by some subterranean species, contribute to accumulation of HMM-HA. We also

experimentally confirmed that changes in expression levels and amino acid substitutions in these enzymes affect the HMM-HA levels. Therefore, HMM-HA in subterranean species arose not due to a shared gene mutation but evolved convergently in each subterranean lineage via a different set of modifications in HA synthesis and degradation pathways.

An interesting observation in our study is that selection tends to alter HA degradation pathway more than the synthesis pathway. This includes a generally lower expression (Fig. 3b) and mutations/pseudogenization (Fig. 3c–e) of HA degrading genes. The possible reason is that loss-of-function mutations occur much more frequently than gain-of-function mutations during evolution[52]. Therefore, there is a greater chance to acquire compromised HA degradation under the selective pressure.

There are two possible benefits that HMM-HA confers to the subterranean species. It has been hypothesized that the HMM-HA of NMR has evolved to provide elastic skin that facilitates squeezing into underground tunnels[20]. Our study further supports this hypothesis. Natural selection favors those individuals living underground with flexible skin enriched with HMM-HA. This is particularly important for NMR and DMR, the only two eusocial mammalian species[53,54], for which multiple individuals squeeze through the same tunnels daily.

We observed changes in BM genes in subterranean mammals. BM-related proteins laminin-322 and nidogen showed adaptive evolution, which may alter the properties of BM potentially contributing to a stronger and denser BM in the skin. However, denser BM may present a problem for other tissues such as alveolar of the lung and glomerulus

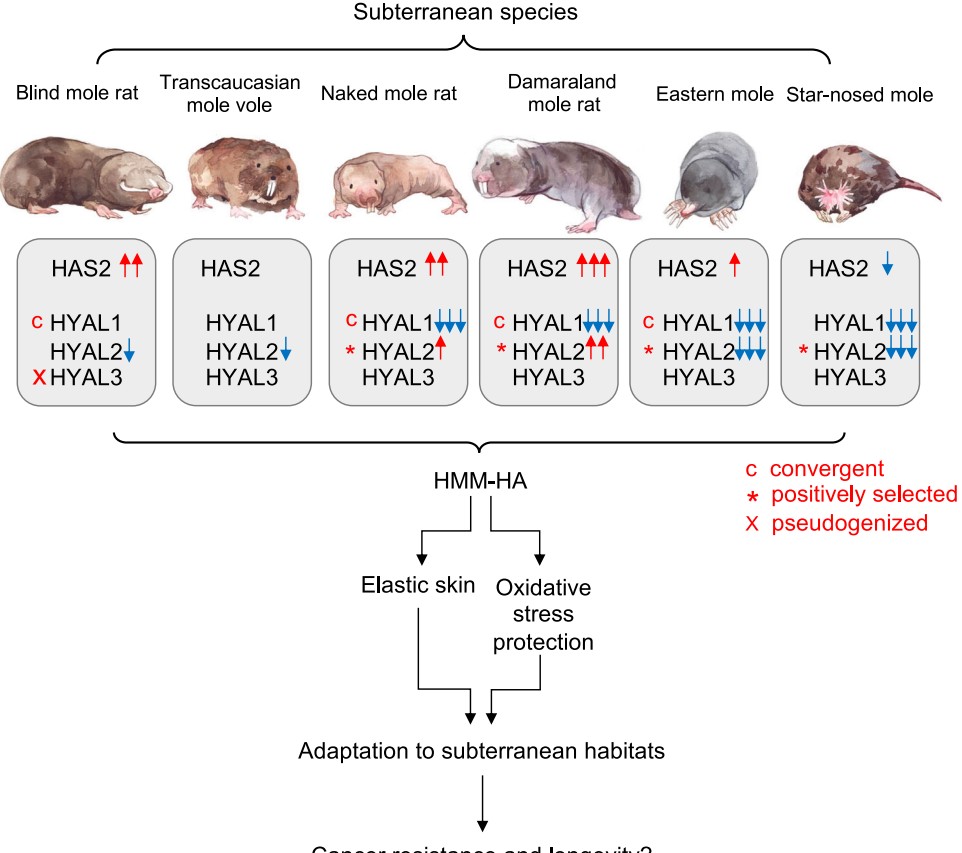

**Fig. 7 | Evolution of HMM-HA in subterranean species.** Subterranean species exhibit high amount of HMM-HA in their tissues. This is achieved by altering the expression of HAS2 and HYAL genes, combined with positively selected or convergent mutations. HMM-HA confers elasticity to the skin and protects from oxidative stress, both contributing to the adaptation to subterranean habitats. This adaptation may be further coopted to provide cancer resistance and longevity to subterranean species. Red and blue arrows indicate higher and lower expression, respectively. c, convergently evolving genes. *, positively selected genes. x, pseudogenization.

of the kidney. Interestingly, we observed different expressional changes of BM collagen genes between skin and other tissues when comparing subterranean and aboveground samples (Supplementary Fig. S11). For example, COL4A1 and COL4A2 expression in subterranean lung and kidney samples were lower compared to aboveground samples, while skin samples showed similar expression. Therefore, the lower expression of major BM collagens in lung and kidney may have evolved as a compensatory mechanism of the increased levels of HMM-HA.

Another major stress of subterranean lifestyle, which can be alleviated by HMM-HA, is hypoxia. With the exception of SNM[55], most of the subterranean species are enduring severe hypoxia and high levels of $CO_2$ in the burrows. The $O_2$ and $CO_2$ levels fluctuate with seasonal changes, different activity of the animals, and the depth of burrows, eventually imposing a great risk of oxidative stress[56–59]. HA has been shown to protect from oxidative stress-induced mitochondrial DNA damage and apoptosis in chondrocytes[60]. Recently, a study showed that NMR HMM-HA has a superior cytoprotective effect against oxidative stress through attenuating p53 signaling[17]. Hence, HMM-HA can help subterranean animals deal with oxidative stress caused by hypoxia.

Perhaps the most striking effect of HMM-HA is its role in cancer resistance in the NMRs[1]. NMR HMM-HA triggers a signaling cascade through the membrane receptor CD44 and induces a p16-dependent early contact inhibition[1]. It has been speculated that HMM-HA of NMRs first evolved as an adaptation to subterranean environments, and later has been co-opted to protect NMRs from cancer[20]. Species living in protected environments evolve longer lifespans due to a lower extrinsic mortality[61]. Subterranean species are exposed to far fewer predators than their aboveground counterparts. Both NMRs and BMRs are exceptionally long-lived for their size and show resistance to cancer[1–3]. As HMM-HA is also a common phenotype shared by subterranean species, it may be of interest to test if HMM-HA contributes to cancer-resistance in all subterranean mammals. While aboveground shrews are among the shortest-lived mammals with the maximum lifespan of 2 years[62], there is little information available about the maximum lifespan of moles, and records indicate EM live over 6 years[63]. Hence, HMM-HA may also contribute to longevity of subterranean species.

In conclusion, our study revealed that HMM-HA is a common phenotype specific to subterranean mammals, contributing to their adaptation to underground habitats. The formation of HMM-HA requires a change in the balance between HA synthesis and degradation. As a result of convergent evolution, subterranean species use different regulatory mechanisms involving both increased synthesis and slower degradation of HA to yield HMM-HA. Our study supports the hypothesis that HMM-HA has evolved as an adaptation to subterranean lifestyle and suggests HMM-HA may contribute to longevity and cancer resistance of a broad range of subterranean mammals.

## Methods

### Animals

Experiments were approved by the University of Rochester Committee on Animal Resources (UCAR). Sources of animals were as previously described[64,65]. Blind mole rats (BMR, *Nannospalax galili*), previously caught in Upper Galilee Mountains in Israel, were maintained in the colonies of University of Rochester. Tissues of Transcaucasian mole vole (TMV, *Ellobius lutescens*) were obtained from Dr. Yuksel Coskun lab at Dicle University, Turkey. Star-nosed moles (SNM, *Condylura cristata*) were collected in New York State. Tissues and cells of eastern moles (EM, *Scalopus aquaticus*) and short-tailed shrews (shrew, *Blarina brevicauda*) were obtained from Dr. Richard Miller lab at University of Michigan. Naked mole rats (NMR, *Heterocephalus glaber*) and Damaraland mole rats (DMR, *Fukomys damarensis*) were from the colonies at University of Rochester. Wild-collected star-nosed moles

are covered by scientific license to collect or possess issued by New York State department of environmental conservation (Scientific #2850). For wild-collected animals, three to five young adults with random sexes were obtained unless indicated otherwise in the Figure legends.

### Cell lines and tissue samples

Skin fibroblasts were freshly isolated from sacrificed shrew, DMR, EM, and SNM by digesting in DMEM/F12 media with 0.14 Wunsch units/mL Liberase Blendzyme 3[66]. Fibroblasts from a variety of other mammalian species, including mouse, rat, guinea pig (GP), golden hamster, beaver, capybara, woodchuck, eastern gray squirrel, red squirrel, chinchilla, Mongolian gerbil, African spiny mouse, degu, shrew, BMR, and NMR, were isolated previously[64,65]. Cells were grown in complete media containing Eagle's Minimum Essential Medium (EMEM, ATCC), 15% (vol/vol) fetal bovine serum (FBS, Gibco), 100 U/mL Penicillin and 100 µg/mL Streptomycin (Gibco). NMR cells were maintained at 32 °C with 5% $CO_2$ and 3% $O_2$. All other fibroblasts were cultured at 37 °C with 5% $CO_2$ and 3% $O_2$. HeLa and HEK293T cells were purchased from ATCC and cultured in DMEM (Gibco) containing 10% FBS, 100 U/mL Penicillin and 100 µg/mL Streptomycin, and were maintained at 37 °C with 5% $CO_2$ and ambient $O_2$. Skin, heart, and kidney from the aforementioned species were collected and frozen in −80 °C freezer or fixed in 4% Paraformaldehyde (PFA).

### Media viscosity assay

For conditioned media, cells were allowed to grow until confluent, media was changed and maintained for additional 10 days before collecting for viscosity assay. For media viscosity, 3 mL of distilled $H_2O$, unused complete EMEM media, or conditioned media of cell lines from different species were run through a 0.6-mm capillary Ostwald viscometer (Barnstead International) at 22 °C. The times of passage of the media or distilled $H_2O$ were recorded. The relative viscosity of each media is determined by the time required to pass through the capillary relative to that of $H_2O$. Samples were run three times to ensure the reproductivity. For each species, cell lines from at least two animals were tested to confirm the reproducibility. For hyaluronidase (HAase) treatment, HAase from *Streptomyces hyalurolyticus* (Sigma-Aldrich) was added to the media to a final concentration of 1 U/mL and incubated overnight at 37 °C.

### Hyaluronic acid (HA) extraction and electrophoresis

Media conditioned with each cell line for 10 days was collected for HA extraction. A 1/10 volume of proteinase K (1 mg/mL) in proteinase K buffer (10 mM Tris-Cl pH8.0, 25 mM EDTA, 100 mM NaCl,0.5% SDS) was added to the media, and incubate at 55 °C for 4 h with rotation. An equal volume of Phenol: Chloroform: Isoamyl Alcohol (Thermo Fisher) was mixed with the media and centrifuged with a swing rotor at 2000 × g for 10 min at 4 °C. The supernatant aquatic phase was transferred to a new tube. After adding 2.5 volumes of pure ethanol and mixing, the tubes were centrifuged at 2500 × g for 1 h at 4 °C. The pellet was washed with 70% ethanol by centrifuging at 2500 × g for 10 min, and then dissolved in 2 mL PBS overnight at room temperature. The next day, the samples were pre-cooled on ice, and 1/100 volume of Triton X-114 was added. After vortexing for 5 min, the samples were incubated at 37 °C for 5 min and centrifuge at 12,000 × g for 10 min at 37 °C. Supernatants were collected to new tubes and NaCl (final 0.15 M) and 2.5 volumes of ethanol were added. Samples were then mixed and centrifuged at 10,000 × g for 15 min at 4 °C. Pellets were washed with 70% ethanol, air-dried, and dissolved in 500 µL PBS.

For electrophoresis, both regular agarose gel and pulsed-field gel were performed. For pulsed-field gel electrophoresis, 25 µL of each sample was mixed with 5 µl 4 M sucrose loading solution and loaded to a 0.4% pulsed-field SeaKem Gold agarose gel (Cambrex). The sizes of HA samples were determined by loading 10 µL of HA molecular size

ladders, including MegaLadder (2000 kDa to 8000 kDa), HiLadder (500 kDa to 1500 kDa), and LoLadder (30 kDa to 500 kDa). Samples were run at 4 V for 16 h with a 1–10 running ratio in TBE buffer using a CHEF-DRII system (Bio-Rad). For agarose gel, 5 or 10 μL of each sample was mixed with 4 M sucrose and loaded to a 0.8% Seakem Gold Agarose/TBE gel. Samples were run at 23 V for 0.5 h followed by 35 V for 6 h. For both methods, the gel was stained by 0.005% (w/v) Stains-All (Sigma-Aldrich) in 50% ethanol overnight. The gel was then washed with distilled $H_2O$ with 10% ethanol for 2 h. The gel was either photographed under white light or captured in a gel imager (Bio-Rad) using the Coomassie blue mode. The intensities of HA lanes were quantified using ImageJ.

### Carbazole assay

For quantification of HA, samples were broken down into HA monomers by mixing 25 μL purified HA with 100 μL of 25 mM sodium tetraborate in sulfuric acid. Mixed samples were heated in boiling water for 10 min, and cooled down at room temperature for 15 min. The samples were then added with 50 μL of 0.125% carbazole solution dissolved in pure ethanol and heated in boiling water for 10 min. Samples were then measured for absorbance at 525 nm.

### ELISA

HA ELISA was used to quantify HA from the cell media using a DuoSet Hyaluronan ELISA kit (R&D DY3614) following the manufacture.

### Immunofluorescence

Skin, heart, and kidney were isolated from mouse, rat, GP, NMR, BMR, DMR, TMV, EM, and SNM. Tissues were fixed in 4% paraformaldehyde (PFA) in PBS overnight at 4 °C. The next day, the tissues were desiccated in 30% (w/v) sucrose in PBS for 24–48 h at 4 °C. Desiccated tissues were frozen in 2-methylbutane on dry ice for 1 min and stored in −80 °C freezer or directly subjected to cryosection. The tissues were sectioned into 25 μm slices by using a Leica SM2000R Sliding Microtome sectioning machine. The slices were washed with PBS for three times and were blocked by 5% Normal Goat Serum (NGS) in PBST (PBS + 0.1% tween 20) for 2 h at room temperature. Following blocking, slices were incubated with biotinylated-HABP (Amsbio) in blocker buffer overnight at 4 °C. Different concentrations of HABP were used for different tissue types according to optimization. HABP was diluted to 1:500 for heart, 1:200 for skin, and 1:100 for kidney. For each sample, a negative control was included by HAase treatment. For HAase treatment, samples were incubated with 1 U/mL HAase dissolved in 100 mM ammonium acetate for overnight at 37 °C. After HABP incubation, slices were washed with PBST for three times followed by incubating with streptavidin-Alexa fluor 647 (1:1000, Invitrogen) in dark for 2 h at room temperature. After washing with PBST for three times and PBS for one time, the slices were incubated with 2 μM Hoechst solution for 30 min and mounted on microscope slides with cover glasses. The samples were detected by using a Nikon Eclipse Ti-S inverted fluorescence microscope. For quantification, similar regions of each organ were selected by controlling cell density using Hoechst staining. At least three such fields of each sample were randomly captured for quantification of fluorescence signals. The average intensities of HA signals were quantified using ImageJ, normalized to the area of selected regions. Experiment was repeated from at least two animals of each species to confirm the reproducibility.

### Quantitative reverse transcription PCR (RT-qPCR)

Total RNA was extracted from cells or tissues using a PureLink RNA Mini Kit (Thermo Fisher), with DNase I on column digestion to remove genomic DNA contamination. To avoid the influences of cell confluency on gene expression, all the cells were kept confluent for 10 days before harvesting for RT-qPCR. cDNA was generated using an iScript cDNA Synthesis Kit (Bio-Rad). cDNA amplification was

performed using SYBR Green supermixes (Bio-Rad). Quantitative PCR was performed using a CFX Connect real-time PCR detection system (Bio-Rad). The relative RNA levels of interested genes were normalized to β-actin. The sequences of primers: for mouse, BMR, and TMV: bmrHAS2-forward: 5′-CTCTGGGAATGTACAGAAACTC-3′; bmrHAS2-reverse: 5′-CGTAGGTCATCCACAAGTGATG-3′; bmrHYAL1-forward: CATGCCTGAACCTGACTTCT; bmrHYAL1-reverse: GTAGCAGTCAGG-GAAGCCATA; bmrHYAL2-forward: 5′-CACCTGCCCATGCTGAAGGA-3′; bmrHYAL2-reverse: 5′-TCAGGAAAGAGGTAGAAGCC-3′; for GP, NMR, and DMR: nmrHAS2-forward: 5′-GGAGTCTCTCTTCTCCTTG-3′; nmrHAS2-reverse: 5′-GTAGGTGAGCCTTTTCACAG-3′; nmrHYAL1-forward: AGTCTACCAGTGCTGCCCTA; nmrHYAL1-reverse: CACTGGT-CACATTCAGGATG; nmrHYAL2-forward: 5′-ATGTGTATCGCCAGT GGTC-3′; nmrHYAL2-R: 5′-AAGAGGTAGAAGCCCCAGA-3′; for shrew, EM, and SNM: moleHAS2-forward: 5′-GGGTGGAAAAAGAGAAGTC-3′; moleHAS2-reverse: 5′-TGAGGAAGGAGATCCAGGA-3′; moleHYAL1-forward: GTGGATGTGGATGTCAGTG; moleHYAL1-reverse: GGCAGGC-CACCAAACAC; moleHYAL2-forward: 5′-CTTTGTGGTAGCATGGGA-3′; moleHYAL2-reverse: 5′-CTGGTTCACAAAACCCTC-3′.

### Plasmids, mutagenesis, and transfection

For overexpression, the coding sequences (CDS) of *HAS2*, *HYAL1*, and *HYAL2* genes from mouse, BMR, Shrew, and SNM as well as *HYAL2* of NMR were cloned into a piggyBac vector. A Kozak sequence was introduced before the CDS to enhance the translation efficiency. The mutant plasmids including SNM HYAL2 H106L, N280H, K284A, S348T, A352V and NMR HYAL2 G284A were constructed using a site-directed mutagenesis kit. For transfection, $2 \times 10^6$ HeLa or HEK293T cells were seeded to 10 cm culture dishes. The next day, the cells were transfected with 5 μg of plasmid using a PolyJet transfection reagent (SignaGen). Media was changed 5 h after transfection. For HYAL1 and HYAL2 degradation assays, transfected cells were incubated with fresh media containing 20 μg/mL (for HEK293T cells) or 10 μg/mL (for HeLa cells) commercial high molecular weight hyaluronan (R&D GLR002). Two days after incubation, conditioned media was collected and HA was purified for electrophoresis.

### RNA sequencing (RNAseq)

Total RNA was extracted from skin, heart, and kidney of mouse, rat, GP, NMR, BMR, DMR, EMV, EM, and SNM by using a Monarch Total RNA Miniprep Kit (New England BioLabs), with DNase I on column digestion to remove genomic DNA contamination. The RNAseq libraries were generated with the Illumina TruSeq stranded total RNA RiboZero Gold kit and sequenced with Illumina NovaSeq 6000 paired-end 150 bp sequencing at University of Rochester Genomics Research Center.

### Transcriptome assembly, annotation and gene expression analysis

Raw reads were demultiplexed using configurebcl2fastq.pl (v1.8.4). Adapter sequences and low-quality base calls (threshold: Phred quality score <20) in the RNAseq reads were first trimmed using Trim_Galore (v0.6.6) via calling the Cutadapt tool[67]. For all species, the clean reads were aligned using Salmon (v1.5.1) to longest transcript of each gene extracted from the de novo assembled transcriptome using Trinity (k-mer=25, --SS_lib_type RF; v2.4.0)[68]. For species with assembled transcriptomes, TransDecoder (https://github.com/TransDecoder/TransDecoder/) (v5.5.0) were used to predict open reading frame and corresponding peptides. The orthologous transcripts of each species to mouse reference were identified by performing reciprocal blast search (BLAST+ v2.10.1; blastp)[69] against human longest protein (GRCh38.p13; Ensembl database, release109) with parameters of "-evalue 1e-05; -max_target_seqs 1" and hits with query coverage > 30% were retained. The values of TPM, read count and effective gene length for each transcript were collected and integrated into transcript-sample table for

each tissue according to their orthologous relationship. The count tables were used for downstream analysis including normalization, differentially gene expression analysis and tissue bias estimation.

Salmon transcript counts were used to perform differential expression analysis. For each tissue count matrix, to filter out genes expressed only in small number of species, only transcripts expressed (with detected reads) in at least two third of all samples with mean read counts >10 were retained. The filtered count matrix was normalized using median of ratios method[70] implemented in DESeq2 package[71]. Since orthologous transcript lengths could vary among different species, we implemented an additional length normalization step in the DESeq2 pipeline to avoid biased comparative quantifications resulting from species-specific transcript length variation. To do this, the matrix of effective transcript lengths for each transcript in each sample was delivered to the DESeq2 'DESeqDataSet' object so that they are included in the normalization for downstream analysis. The count matrix of all orthologous transcripts among all samples was normalized in the same way and was used to perform hierarchical clustering. The normalized expression data were passed though "removeBatchEffect" function in the limma R package[72] to remove potential batch effects. Differential expression analysis was performed using the Wilcoxon rank-sum test with *wilcox.test* function in R (v4.1.0) in order to reduce the false positives that may be generated using parametric methods when the sample size is large. This method was shown to have solid FDR control and good power[73]. We considered all differentially expressed genes (DEGs) with an adjusted $p < 0.01$, and 1.5-fold median expression change to be statistically significant in our analyses.

## Functional enrichment analysis

Gene ontology (GO) over-representation and GSEA analysis were performed by R package clusterProfiler (v3.14)[74]. For over-representation analysis, the DEGs identified from each tissue were used to perform GO enrichment analysis. All three orthogonal ontologies, including molecular function (MF), biological process (BP), and cellular component (CC) were considered. All human genes in 'org.Hs.eg.db' were used as background gene list. Fisher's exact test was performed for each GO term and corresponding $p$-value was calculated. All the Fisher's exact test $p$-values were adjusted for multiple testing with Benjamini-Hochberg (BH) correction. To take more GO terms into account, GO terms with $p$-value < 0.05 were considered enriched. For GSEA analysis, the ranked fold changes from the DEG analysis were used. Similarly, all orthogonal ontologies were included. The $p$-values are calculated using a permutation-based approach. Alternatively, we used the General Application Gene-set Enrichment (GAGE) package in R[48] to identify GO terms with the most consistent differences in gene expression across subterranean lineages compared to aboveground lineages. For each tissue, we further quantile-normalized the DESeq2 normalized expression matrix and used them as the input of GAGE. Samples from subterranean lineages were always marked as 'target' and samples from aboveground lineages as 'reference', and we fixed the 'compare' option as unpaired when comparing differences in gene expression between subterranean and aboveground replicates. The custom ECM gene sets (see "ECM-related genes compilation" section) GSEA analysis follows the same procedure to GO terms GSEA analysis.

## ECM-related genes compilation

To compile a gene set that may cover or be involved in the majority of the ECM components, we considered several sources including public database, literatures and gene ontology. The primary source of ECM-related genes is from Matrisome project (http://matrisomeproject.mit.edu; Human Matrisome updated August 2014)[75], which contains core ECM proteins such as collagens, glycoproteins and proteoglycans. Because glycosyltransferases and glycoside hydrolase play an important roles in ECM assembly and remodeling, two sets of genes encoding these enzymes were included[76,77]. To further expand the

catalogs, we also searched the ECM-related GO terms (http://geneontology.org) using keywords including "extracellular matrix", "hyaluronan", "hyaluronic", "chondroitin sulfate", "heparan sulfate", "dermatan sulfate" and "keratin sulfate".

All lists were merged together, and the redundant genes were removed. As a result, we obtained 2932 ECM-related genes in total.

## Alignment construction

Multi-sequence alignments were constructed based on a phylogenetic tree including 17 species (Supplementary Data 1) and 15 species (Supplementary Data 2, without zokor and bamboo rat) using elephant and armadillo as outgroup. The following public genomic data were used to extract the coding sequences (CDSs) for constructing multi-sequence alignments: human (GRCh38.p13), mouse (GRCm39), rat (mRatBN7.2), BMR (S.galili_v1.0), plateau zokor (plateau_zokor_v2.1), hoary bamboo rat (RhiPru_1.0), NMR (HetGla_female_1.0), DMR (DMR_v1.0), kangaroo rat (Dord_2.0), GP (Cavpor3.0), dog (ROS_Cfam_1.0), elephant (Loxafr3.0), armadillo (Dasnov3.0) genome assemblies were obtained from the Ensembl or CNCB database. CDSs of TMV, EM, SNM and shrew were collected from the assembled transcriptomes. Human-referenced orthologous groups were constructed based on the reciprocal best alignments with human. Before alignment, sequences were filtered based on the following criteria: (i) sequence with low quality (e.g., >50% missing data) were removed; (ii) sequences containing internal stop codons or sequences <150 nt (likely to cause alignment errors) were removed; (iii) removing certain number of nucleotides at the end of sequence to ensure the sequences are triple. Sequences were then aligned with GUIDANCE (v.2.02)[78] and PRANK[79] with codons enforced and 10 bootstraps. Aligned sequences were further filtered with following criteria: (i) sequence with a GUIDANCE score <0.6 were removed, and the alignment process was repeated; (ii) codons with any site having column score <0.93, and/or >50% gaps, were removed with a Perl script; (iii) Sequences <150 nt (50 codons) were removed, and only alignments >100 codons were kept. (iv) Only alignments containing ≥2 subterranean species were kept. As a result, 2497 multi-sequence alignments were retained for downstream analysis.

## Testing for positively selected genes (PSGs)

Each gene was tested for positive selection along the branches leading to each subterranean species (exterior nodes) or the most recent common ancestor (interior nodes) for NMR/DMR and EM/SNM. We implemented branch-site models (optimized Model A) with codeML in PAML (v.4.8)[30] to test for positive selection along each of the branch of interests. We repeated similar selection tests on four terrestrial control branches, which are close sister taxa of each subterranean lineages. Site-wise estimates of ratio of nonsynonymous to synonymous rates (dN/dS; $\omega$) were estimated separately for the branch of interest (designated as foreground branch) and across the remaining background branches. Four classes of sites are considered: $0 < \omega_0 < 1$, $\omega_1 = 1$, $\omega_{2a}$ can exceed 1 on the foreground but less than 1 on the background, and $\omega_{2b}$ can exceed 1 on the foreground but equals to 1 on the background. Likelihood ratio test (Test 2) was implemented in which we compared the modified model A (alternative model) with the corresponding null model with $\omega_2 = 1$ fixed. The null distribution was a 50: 50 mixture of a chi-squared distribution with 1 degree of freedom and a point mass at zero. The significance of model fit was assessed by chi-square test with $P < 0.05$ indicates a significantly better fit of the alternative model. To address the multiple testing issue, we reported FDR for each gene. Genes with FDR < 0.1 was considered as PSGs. We further filtered the PSGs list to reduce the potential false positives caused by alignment errors according to a method used by previous studies[4,80,81]. Briefly, genes with a median interval between positively selected sites ≤10 amino acids were removed.

## Detecting convergent molecular evolution in ECM-related genes

The coding sequences for each orthologous gene in our data set was aligned using PRANK (45) and further filtered (see "alignment construction" section). The filtered CDS alignments were translated into amino acid sequence alignments. On the basis of the same phylogenetic tree used in PSGs detection, we inferred ancestral sequences (all interior nodes) for each gene using maximum likelihood and empirical Bayesian approaches[30], counting the numbers of convergent and divergent substitutions along the branch pairs that belong to either subterranean or aboveground lineages. Corrected maximum likelihood estimates of parameters, such as the proportions and dN/dS ratios for the site classes, were used as the priors. For any pair of species, the extant sequences at each position were compared to the ancestral sequence at the node corresponding to their most recent ancestor. To reduce the false positives introduced by sequencing errors, incorrect alignments, and non-orthologous regions in the alignments, we deleted a convergent/divergent site if its ±10 amino acids flanking sequences met one of the following criteria: (i) mean sequence similarity <0.7; (ii) lowest similarity <0.35 between any two sequences; and (iii) >2 successive indels in more than two species. The convergent substitutions in this study include both convergent and parallel substitutions that resulted in the same amino acid along the branch pairs examined. If a gene contained at least one convergent substitution, then it was defined as a convergent gene.

## Detection of gene loss and relaxed selection

In this study, gene loss defined as genes harboring ORF-disrupting mutations, including premature stop codons and/or frameshifts with intact 1:1 orthologs in the human genome. The gene loss identification procedure follows a previous study[31]. Briefly, The longest protein of human HA related genes including HAS1-3 and HYAL1-3 were used to identify 1:1 orthologs in each subterranean species using genBlastA[82]. The orthologous genomic sequences with 5000 bp up and downstream flanking sequences were extracted to predict the gene structure and ORF using GeneWise[83], disruptive mutations were identified using custom Perl script. RELAX[84] were used to detect relaxed selection constraint signals on BMR HYAL3. The CDS of GeneWise alignment (with disruptive mutations removed) were extracted and aligned with orthologous CDSs from all aboveground species in the phylogeny used in PSG analysis, so that only a single subterranean species was contained in the alignment. PRANK was used to construct codon-based multiple sequence alignment. Gaps were manually removed. In the alignment, BMR was set as "Foreground" branch and the rest of species were set as "Background" branch. The selection intensity parameter $k > 1$ indicates intensified selection and $k < 1$ indicates relaxed selection.

## Structural analysis of HYAL2 protein

Structural comparisons were made by using model HYAL2 structures. A human HYAL2 model was generated by AlphaFold (29) and EM and SNM HYAL2 models were generated using SWISS-MODEL using human HYAL1 (PDB code: 2PE4) as a template. The HA tetramer was mapped using an aligned bee venom HYAL structure (PDB code: 1FCV). The structure models were validated by aligning to human HYAL1 (PDB code: 2PE4) and bee venom HYAL (PDB code: 1FCQ + 1FCV). All structures align well with all alignments having RMSD's below 1.5 and justifies the use of the generated HYAL2 structures[34,85–95]. To generate figures, EM and SNM HYAL2 models were aligned to the human HYAL2 AlphaFold model using PyMOL[33].

## Tissue bias analysis for PSGs and CEGs

The tissue bias estimates were calculated for all expression genes for all species with available expression data. The mean expression (quantile-normalized TPM) of all samples of each tissue were used to calculate tissue specificity index $\tau$[47] using "calcTau" function from "roonysgalbi/tispec" R package (https://rdrr.io/github/roonysgalbi/

tispec/src/R/calcTau.R). Genes with $\tau > 0.25$ were identified as tissue-specific genes. The density plot of tissue bias for ECM-related PSGs or CEGs that pooled from two independent analysis and non-PSGs were plotted in each species (represented by two subterranean species and one control species) using "plotDensity" function from "roonysgalbi/tispec" R package.

## Statistical information

Error bars show standard deviation (SD). Two-tailed Student's $t$ test was used to test the statistical significance of the differences between groups unless otherwise indicated. $P < 0.05$ was set as a threshold for statistical significance.

## Reporting summary

Further information on research design is available in the Nature Portfolio Reporting Summary linked to this article.

## Data availability

Most data are included in the figures. RNAseq data used in this study are available in the Gene Expression Omnibus (GEO) under accession no. GSE181413 and GSE190756. Genomic data used in this study are available through the following accession numbers and links: human (GRCh38.p13; GCF_000001405.39, https://ftp.ncbi.nlm.nih.gov/genomes/all/GCF/000/001/405/GCF_000001405.39_GRCh38.p13), mouse (GRCm39; GCF_000001635.27, https://ftp.ncbi.nlm.nih.gov/genomes/all/GCF/000/001/635/GCF_000001635.27_GRCm39), rat (mRatBN7.2; GCF_015227675.2, https://ftp.ncbi.nlm.nih.gov/genomes/all/GCF/015/227/675/GCF_015227675.2_mRatBN7.2), BMR (S.galili_v1.0; GCF_000622305.1, https://ftp.ncbi.nlm.nih.gov/genomes/all/GCF/000/622/305/GCF_000622305.1_S.galili_v1.0), plateau zokor (plateau_zokor_v2.1; GWHABJZ00000000, https://ngdc.cncb.ac.cn/gwh/Assembly/941/show), hoary bamboo rat (RhiPru_1.0; GCA_009823505.1, https://ftp.ncbi.nlm.nih.gov/genomes/all/GCA/009/823/505/GCA_009823505.1_RhiPru_1.0), NMR (HetGla_female_1.0; GCF_000247695.1, https://ftp.ncbi.nlm.nih.gov/genomes/all/GCF/000/247/695/GCF_000247695.1_HetGla_female_1.0), DMR (DMR_v1.0; GCF_000743615.1, https://ftp.ncbi.nlm.nih.gov/genomes/all/GCF/000/743/615/GCF_000743615.1_DMR_v1.0), kangaroo rat (Dord_2.0; GCF_000151885.1, https://ftp.ncbi.nlm.nih.gov/genomes/all/GCF/000/151/885/GCF_000151885.1_Dord_2.0), GP (Cavpor3.0; GCF_000151735.1, https://ftp.ncbi.nlm.nih.gov/genomes/all/GCF/000/151/735/GCF_000151735.1_Cavpor3.0), dog (ROS_Cfam_1.0; GCF_014441545.1, https://ftp.ncbi.nlm.nih.gov/genomes/all/GCF/014/441/545/GCF_014441545.1_ROS_Cfam_1.0), elephant (Loxafr3.0; GCF_000001905.1, https://ftp.ncbi.nlm.nih.gov/genomes/all/GCF/000/001/905/GCF_000001905.1_Loxafr3.0), armadillo (Dasnov3.0; GCA_000208655.2, https://ftp.ncbi.nlm.nih.gov/genomes/all/GCA/000/208/655/GCA_000208655.2_Dasnov3.0). The data generated in this study are provided in the Source Data file. The exact $P$ values, if applicable, are included in the paper and in the Source Data. Source data are provided with this paper.

## Code availability

The custom Perl script used in this study will be available upon request.

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

## Acknowledgements
We thank Wendy R. Hood and Geoffrey E. Hill for providing shrew samples. We thank Ms. Ke Guo for drawing the animals of Fig. 7. This research was supported by US National Institute on Aging grants to V.G. (grant no. AG047200) and A.S. (grant no. AG047200), and by Zhejiang Provincial Natural Science Foundation of China to Y.Z. (grant no. LZ23C110002).

## Author contributions
Y.Z., A.S. and V.G. designed the research; Y.Z., Z.Zha., Y.X., Y.S.L., F.T.Z. and S.A.B performed experimental research; Z.Zhe., E.H., J.Y.L., Y.Z., A.S. and V.G. analyzed data; J.A. and R.A.M. provided samples for the research; and Y.Z., Z.Zhe., F.T.Z., E.N., A.S. and V.G. wrote the paper.

## Competing interests
The authors declare no competing interest.
