## [Peer Review File · Nature Communications]

Evolution of High-Molecular-Mass Hyaluronic Acid is Associated with Subterranean LifestyleREVIEWER COMMENTS

Reviewer #1 (Remarks to the Author):

Overall comments: In this interesting study the authors explore the existence of HMM-HA across numerous subterranean species to determine if this adaptation is shared among such animals. I enjoyed reading the manuscript and applaud the authors for their multi-species approach! The results convincingly demonstrate that subterranean rodents and moles all express HMM-HA. The authors then use PCR approaches to extensively explore regulatory components of HMM-HA function in comparison to surface-dwelling species and seek signals of convergent evolution. The genetic work is promising but does not constitute a sufficiently in-depth mechanistic study to draw meaningful conclusions regarding the regulation of these systems across species. I recommend that the authors conduct additional studies at the protein/enzyme level to test some of the intriguing hypotheses that they put forward regarding the control of HMM-HA control across their many fascinating study species.

Major comments:

- Abstract, intro and conclusions: The authors repeatedly suggest that HMM-HA in subterranean species contributes to their cancer resistance and longevity, but this link has only been demonstrated in NMRs and there are many mechanisms that have been shown to contribute to the cancer resistance of NMRs in particular. I think this statement is a bit of an overreach of the data and unless the authors have data to show cancer resistance or longevity-associated benefits in the other subterranean species then this comment should be removed from the abstract and introduction but kept in the discussion where more context is provided.

- It is good that the authors measured HAS2, HYAL2, and HYAL1 gene expression; however, transcription of genes is only a small part of the regulation of the activity of these enzymes. It would be a reasonable addition to include protein/enzyme assay measures to assess their actual function. In particular, the authors highlight compromised degradation pathway activity in subterranean species, which could be functionally tested in cells/tissues.

- The authors go on to propose several mutations to HYAL2 that they suggest impacts their function but unfortunately do not test them. To publish in a high-impact journal such as this one I would expect further mechanistic study of the functional molecular differences that differentiate the regulation of HMM-HA in surface vs. subterranean species that goes well beyond PCR analysis.

- Statistical information is not provided anywhere. Please specify your methods of statistical analysis, including test and post-test information, alpha value, etc. Please provide f-stats where appropriate in the results section.

Minor comments:

- The manuscript is generally well-written but there are numerous grammatical and syntax errors and I encourage the authors to proof-read their submission more carefully. Some examples (there are several others):

- o Abstract: authors use both HM weight and HM mass to describe HMM-HA
- o ECM is defined 2x in the first intro paragraph
- o Lines 66 and 68: the ECM
- o Line 67: delete comma

o Line 79: "superior...than" is not grammatically correct

- Abbreviations are used inconsistently: e.g., lines 111-112 contains long-form of some species and short form of others, even though abbreviations for all study species are introduced in the preceding paragraph. Same comment elsewhere, including in the methods.

- Latin names are inconsistently provided (for all species except mouse and guinea pig)

- The authors refer to differences between gene expression in different species as being up or downregulated but this is somewhat misleading. These terms usually apply to changes in gene expression induced by a treatment, and not to differences between species at baseline. I suggest stating that some species had higher expression or higher endogenous expression instead.

- Lines 366-373: Not all subterranean species are hypoxia tolerant. Star-nosed moles are quite intolerant of hypoxic stress (see Devereaux et al, JEB 2021).

- Methods: provide animal numbers, sex, mass, and age.

- Methods: Provide methods for obtaining fibroblast. Were animals anesthetized? Were they sacrificed? Please provide animal ethics approval information.

- Methods: the body temperature of star-nosed moles is 35C

Matthew Pamerter

Reviewer #2 (Remarks to the Author):

This manuscript builds on previous studies from this group (ref 1) demonstrating that hyaluronan is increased in abundance and size in the naked mole rat as well as work from other groups (Ref 54 and Del Marmol et al [2021], Scientific Reports 11:7951) that examined the sequence and/or expression of genes involved in HA metabolism in the naked mole rat and other subterranean species compared to overland mammals.

Major Comments

1. This work expands the species demonstrated to have increased size and abundance of hyaluronan to include the Damaraland mole rat (DMR), Star-nosed mole (SNM), the Eastern mole (EM) and increased abundance to the Transcaucasian mole vole (TMV). This strengthens the hypothesis that increased abundance and sized hyaluronan is associated with a subterranean lifestyle and therefore would be of interest to groups working on the evolution of subterranean mammals as well as to researchers working in the hyaluronan field with interests in the metabolism of this extracellular molecule.

2. The authors extensively explore the gene expression profiles associated with a range of genes involved in HA metabolism as well as genes encoding protein constituents of the extracellular matrix. This is the first work to explore this broad range of genes in the context of multiple subterranean species. They also analyze the evolution and predicted functional impact of variants present in the hyaluronan-degrading enzymes HYAL1 and HYAL2, work that has not been done by any other groups. However, these studies could have been better described in the context of pre-existing work that did a thorough study of HAS2 evolutionary changes in 11 subterranean mammals and 57 other mammals (ref 54) as well as a study not referenced herein and mentioned above (Del Marmol et al) that provided additional evidence for elevated hyaluronan that was increased in size in the NMR (although not considered to have the high molecular mass suggested by this group) and showed that the only

significant changes in hyaluronan-related genes in the NMR were increased expression of HYAL3 and TNFAIP6 genes.

3. The authors claim that abundant HMM-HA is found in a wide variety of subterranean mammalian species but not in phylogenetically related aboveground species. This argument could be strengthened by expanding the range of both subterranean and aboveground species that were examined.

Compared to the study of HAS2 in moles (Ref 54) this study was relatively small.

4. The second claim that these species accumulate abundant HMM-HA by regulating the expression of genes involved in HA degradation and synthesis and contain unique mutations in these genes is not strongly supported by the data in the manuscript. There were no consistent changes in the HAS2/HYAL2/HYAL1 genes of the subterranean mammals leading the authors to speculate that each species increases HA through a different mechanism. While this is possible, it is very speculative as HYAL1 works intracellularly and in mouse its complete loss can be completely compensated for by other exoglycosidases beta-glucuronidase and beta-hexosaminidase (Gushulak et al. [2012] 11:287:16689-97). Decreased HYAL2 or increased HAS2 activity are better candidates as are some of the other proteins, CEMIP and TMEM2. Overall the large variation in expression of these genes between biological replicates and between species made it difficult to come to any conclusion from these studies. The increased hyaluronan detected in the subterranean species may be completely unrelated to the expression of these genes and instead depend on the expression for example of a hyaluronan-binding protein.

5. The analysis of hyaluronan levels in tissues (skin, heart, kidney) from the subterranean species using quantitative immunofluorescence was interesting but again would benefit from more structured analysis. Some of the things that made this less convincing were that the areas of the sections from within the tissues did not appear to be matched. The levels of hyaluronan vary considerably between different regions in tissues (see de Marmol ref and many others in the field). Therefore it is critical that regions of tissues are matched in the sections and that equivalent areas are quantified. These experiments would have benefitted by comparison to some kind of endogenous control that was considered to be unchanged. This data if carefully done could further substantiate the first claim that hyaluronan is increased in subterranean species.

6. The RNA seq analysis was an exciting addition to this study as it included 6 subterranean species and 4 aboveground species across 6 tissues. The inclusion of a broader range of genes involved in hyaluronan metabolism and function was great as HYAL3-6 are not known to have any role in hyaluronan metabolism. HYAL5 is a pseudogene, HYAL4 a chondroitinase, HYAL3 appears inactive, and HYAL6 is not present in all mammals. This work showed that HYAL3 was down-regulated in multiple tissues of subterranean species while the work of Marmol et al showed it to be upregulated up to 129 fold in NMR compared to mice. Given that the expression levels were averaged for the aboveground and subterranean samples in this study it was difficult to try to reconcile this difference and the authors may want to address this directly. Further, are these studies consistent with the qPCR that was done in the first parts of the study on HAS2, HYAL1 and HYAL2 in specific tissues?

7. The analysis of variants in HYAL1 and HYAL2 was interesting and the authors did strive to predict the impact of these changes by analyzing how these changes would affect the structure of HYAL2 in a model of HYAL2 generated with AlphaFold. These studies would need to be followed up by some in vitro assays of HYAL2 harboring these variants. The size of hyaluronan in the medium of cells transfected with wild type vs these variant-containing forms of HYAL2 would help to provide additional insight into the significance of these variants in the context of HYAL2 activity.

8. The last section of the paper looking at ECM related genes is extremely interesting but felt disconnected from the focus of the paper. It does make sense to go this direction given the lack of a clear linkage with HA metabolic enzymes, but the authors could improve the linkages. This section of the paper seemed to move more in the direction of looking at ECM components as a function of evolution of subterranean species. One apparent link comes from the study of Del Marmol where the TSG-encoding gene is differentially expressed in the NMR compared to the guinea pig. Was this finding reproduced in these studies?

9. The methods sections were detailed enough to allow the experiments to be reproduced.

Minor Comments:

1. The authors sometimes missed the "d" on the star-nosed mole

2. On line 82, the authors describe the size of the NMR HA as extensive; perhaps this should be large rather than extensive which has a different meaning.
3. The authors should keep in mind that some of these genes are affected by the confluency of the fibroblasts (HYAL1 for example, BBRC 266:268-273)
4. The authors did not comment on the limitations of the carbazole assay which will also react with sugars present in the tissue or derived from other polysaccharides. They might consider using an assay specific to hyaluronan such as an HA ELISA (<https://www.echelon-inc.com/product/hyaluronan-enzyme-linked-immunosorbent-assay-ha-elisa/>)
5. The authors should indicate how they normalized the qPCR for RNA levels in these various species.
6. The authors should indicate if all known hyaluronan binding proteins were included in their ECM related genes.
7. The authors do not indicate the sex of the animals that were used in these studies or describe any known sex-specific differences in hyaluronan tissue levels.

Reviewer #3 (Remarks to the Author):

In the manuscript, Zhao et al. reported that the subterranean mammalian species from multiple independent lineages generally contained abundant high-molecular-mass hyaluronic acid (HMM-HA) which has been proven to contribute to the cancer resistance of the naked mole-rat (Tian et al. 2013, Nature), suggesting a general role of HMM-HA in adapting to the underground lifestyle. The authors next performed several evolutionary analyses on the genes involved in HA degradation and synthesis, and found strong signals of the adaptation in these genes, further suggesting the abundant HMM-HA may benefit the adaptation to subterranean environment.

Overall, the results are very inspiring and would be very interesting to the relevant field. Therefore, I would be happy to recommend acceptance after the authors make some revisions.

Specific comments:

- 1) The authors raised the scientific question that "However, the role of extracellular matrix (ECM) in adaptation to subterranean environment, remains poorly understood" at the end of the first paragraph of Introduction. However, this is not the exact question that the study wanted to address. The authors should clarify the question about the evolution of HMM-HA in adaptation to the subterranean lifestyle of mammals.
- 2) The authors missed the results of evolutionary analyses about the genes involved in HMM-HA degradation and synthesis at the end of Introduction. Please add the related information.
- 3) In the first part of Results, the authors should display the statistical results in the main text, just like in the following parts about evolutionary analyses. This is also true for the RT-qPCR results. In the second paragraph, they only compared the size of HA, not the amount of HA, please modify the first sentence of this paragraph.
- 4) The authors missed the methods about how to test the relaxed selection and how to obtain the sequences. Please add the information.
- 5) When displaying the parallel amino acid replacements (Fig. 3d), it would be better to show these sites according to the species' phylogeny.
- 6) In fact, besides the subterranean mammals the authors mentioned in the study, there were several additional underground mammalian species with available high-quality genome sequences such as bamboo rat (Guo et al. 2021; doi.org/10.24272/j.issn.2095-8137.2021.240) and plateau zokor (Xu et al. 2021; doi.org/10.1038/s41422-021-00517-6). The authors are suggested to include these species in their evolutionary analyses to improve the generality of their main point that the evolution of high-molecular-mass hyaluronic acid is associated with the subterranean lifestyle of mammals.

Response to reviewers

Reviewer #1 (Remarks to the Author):

Overall comments: In this interesting study the authors explore the existence of HMM-HA across numerous subterranean species to determine if this adaptation is shared among such animals. I enjoyed reading the manuscript and applaud the authors for their multi-species approach! The results convincingly demonstrate that subterranean rodents and moles all express HMM-HA. The authors then use PCR approaches to extensively explore regulatory components of HMM-HA function in comparison to surface-dwelling species and seek signals of convergent evolution. The genetic work is promising but does not constitute a sufficiently in-depth mechanistic study to draw meaningful conclusions regarding the regulation of these systems across species. I recommend that the authors conduct additional studies at the protein/enzyme level to test some of the intriguing hypotheses that they put forward regarding the control of HMM-HA control across their many fascinating study species.

Response: We thank the reviewer for the positive and supportive evaluation and for the detailed comments.

Major comments:

- Abstract, intro and conclusions: The authors repeatedly suggest that HMM-HA in subterranean species contributes to their cancer resistance and longevity, but this link has only been demonstrated in NMRs and there are many mechanisms that have been shown to contribute to the cancer resistance of NMRs in particular. I think this statement is a bit of an overreach of the data and unless the authors have data to show cancer resistance or longevity-associated benefits in the other subterranean species then this comment should be removed from the abstract and introduction but kept in the discussion where more context is provided.

Response: We thank the reviewer for pointing this out. We removed this speculation in the abstract and introduction and only kept it in the discussion.

- It is good that the authors measured HAS2, HYAL2, and HYAL1 gene expression; however, transcription of genes is only a small part of the regulation of the activity of these enzymes. It would be a reasonable addition to include protein/enzyme assay measures to assess their actual function. In particular, the authors highlight compromised degradation pathway activity in subterranean species, which could be functionally tested in cells/tissues.

Response: We addressed this question through two lines of experiments. The new figures were included and the figure numbers were highlighted.

The first experiment:

We overexpressed HAS2, HYAL1, or HYAL2 from different species in 293T or HeLa cells and tested their effects on HA synthesis and degradation. We noticed that BMR fibroblasts had a 14-fold higher endogenous expression level of HAS2 than mouse cells (Fig. 1e), therefore, we transfected mouse or BMR HAS2 plasmids in a 14-fold difference to mimic the endogenous expression difference. Two days after transfection, HA was purified from the media and was subjected to electrophoresis. The 293T cells expressing 14-fold higher level of BMR HAS2 yielded more abundant HA (Supplementary Fig. S7A). In HeLa cells, the cells expressing high level of BMR HAS2 generated not only significantly more abundant, but also larger-sized HA than cells expressing low level of mouse HAS2 (Fig. 4a). Strikingly, even when equal amount of mouse and

BMR HAS2 were transfected, the size of HA secreted by BMR HAS2-expressing cells was still larger than that secreted by mouse HAS2-expressing cells (Fig. 4a, lane 4 vs. lane 5). Interestingly, the HAS2 gene is highly conserved between mouse and BMR, with only four amino acids different (Supplementary Fig. S8A). Therefore, this result suggests that the four mutations in BMR HAS2, combined with its higher expression, contributed to the HMM-HA in BMR. Similarly, cells expressing SNM HAS2 also generated larger-sized HA than Shrew HAS2 (Fig. 4b). These results suggest that the different sequences and expression levels of HAS2 at least partly contributed to more abundant HMM-HA in subterranean species.

Supplementary Fig. S7A:

Figure legend: 293T cells were transfected with low amount of mouse HAS2 or high amount of BMR HAS2 (14-fold higher) plasmids. Two days after transfection, HA was purified and run on an agarose gel.

Fig. 4a-b:

Figure legends: **a**, HeLa cells were transfected with low amount of mouse HAS2, high amount of BMR HAS2 (14-fold of mouse HAS2), or high amount (equal to BMR HAS2) mouse HAS2 plasmids. Two days after transfection, HA was purified from media and electrophoresis was performed. **b**, HeLa cells were transfected with equal amount of shrew or SNM HAS2 plasmids for two days. HA was purified and run on an agarose gel.

Supplementary Fig. S8A:

A

```
Mouse   MH CERFLC VLR IIG T T L F G V S L L L G I T A A Y I V G Y Q F I Q T D N Y Y F S F G L Y G A F L A S H L I I Q   60
BMR     MH CERFLC I L R I I G T T L F G V S L L L G I T A A Y I V G Y Q F I Q T D N Y Y F S F G L Y G A F L A S H L I I Q   60
*****:*****

Mouse   S L F A F L E H R K M K K S L E T P I K L N K T V A L C I A A Y Q E D P D Y L R K C L Q S V K R L T Y P G I K V M V I   120
BMR     S L F A F L E H R K M K K S L E T P I K L N K T V A L C I A A Y Q E D P D Y L R K C L Q S V K R L T Y P G I K V M V I   120
*****

Mouse   D G N S D D D L Y M M D I F S E V M G R D K S A T Y I W K N N F H E K G P G E T E E S H K E S S Q H V T Q L V L S N K S   180
BMR     D G N S D D D L Y M M D I F S E V M G R D I S A T Y I W K N N F H E K G P G E T D E S H K E S S Q H V T Q L V L S N K S   180
*****

Mouse   I C I M Q K W G G K R E V M Y T A F R A L G R S V D Y V Q V C D S D T M L D P A S S V E M V K V L E E D P M V G G V G G   240
BMR     V C I M Q K W G G K R E V M Y T A F R A L G R S V D Y V Q V C D S D T M L D P A S S V E M V K V L E E D P M V G G V G G   240
*****

Mouse   D V Q I L N K Y D S W I S F L S S V R Y W M A F N I E R A C Q S Y F G C V Q C I S G P L G M Y R N S L L H E F V E D W Y   300
BMR     D V Q I L N K Y D S W I S F L S S V R Y W M A F N I E R A C Q S Y F G C V Q C I S G P L G M Y R N S L L H E F V E D W Y   300
*****

Mouse   N Q E F M G N Q C S F G D D R H L T N R V L S L G Y A T K Y T A R S K C L T E T P I E Y L R W L N Q Q T R W S K S Y F R   360
BMR     N Q E F M G N Q C S F G D D R H L T N R V L S L G Y A T K Y T A R S K C L T E T P I E Y L R W L N Q Q T R W S K S Y F R   360
*****

Mouse   E W L Y N A M W F H K H L W M T Y E A V I T G F F P F L I A T V I Q L F Y R G K I W N I L L F L L T V Q L V G L I K   420
BMR     E W L Y N A M W F H K H L W M T Y E A V I T G F F P F L I A T V I Q L F Y R G K I W N I L L F L L T V Q L V G L I K   420
*****

Mouse   S S F A S C L R G N I V M V M S L Y S V L Y M S S L L P A K M F A I A T I N K A G W G T S G R K T I V V N F I G L I P   480
BMR     S S F A S C L R G N I V M V M S L Y S V L Y M S S L L P A K M F A I A T I N K A G W G T S G R K T I V V N F I G L I P   480
*****

Mouse   V S W F T I L L G G V I F T I Y K E S K K P F S E S K Q T V L I V G T L I Y A C Y W M L L T L Y V V L I N K C G R R   540
BMR     V S W F T I L L G G V I F T I Y K E S K K P F S E S K Q T V L I V G T L I Y A C Y W M L L T L Y V V L I N K C G R R   540
*****

Mouse   K K G Q Q Y D M V L D V   552
BMR     K K G Q Q Y D M V L D V   552
*****
```

Figure legend: Alignment of mouse and BMR HAS2 protein.

The second experiment:

We next compared the HA degradation abilities between subterranean and aboveground species. Commercial high molecular mass HA was incubated with confluent fibroblasts of five subterranean species or three aboveground control species for two days. After incubation, more HA remained after incubated with cells of subterranean species than aboveground species (Fig. 4c). The result remained the same when the HA was incubated with growing cells (Supplementary Fig. S7B), suggesting that subterranean species' cells had weaker HA-degrading abilities.

We then examined the different HA-degrading abilities of HYAL2 and HYAL1 in subterranean species. We overexpressed either HYAL2 or HYAL1 of mouse, BMR, shrew, or SNM in 293T cells. Commercial HA was incubated with the transfected cells for two days, and HA was extracted and subjected to electrophoresis. The shrew HYAL2 induced more degradation of commercial HA than the SNM HYAL2 (Supplementary Fig. S7C-D), suggesting that the HYAL2 in subterranean species had weaker HA-degrading abilities. We performed the same experiment in HeLa cells. While no differences were observed after two days incubation, when the incubation was extended to three days, significant amount of commercial HA was degraded. Strikingly, mouse and shrew HYAL2 degraded more HA, resulting in smaller size of remaining HA compared with BMR and SNM, respectively (Fig. 4d). As expected, cells expressing HYAL1 of different species did not yield significant difference in HA degradation (Fig. 4d and Supplementary Fig. S7E), probably because HYAL1 is an intracellular enzyme. Taken together, these results suggest that subterranean HYAL2 enzymes had weaker HA-degrading activities, contributing to HMM-HA.

We added these new results in the manuscript.

Fig. 4c-d:

Figure legends: **c**, Media containing 20 µg/mL commercial high molecular weight HA was incubated with confluent fibroblasts from each species. Two days after incubation, HA was purified from the media and run on an agarose gel. For each cell line, a sample without commercial HA was included to control the endogenous HA. **d**, HeLa cells were transfected with either HYAL2 or HYAL1 of mouse, BMR, shrew, or SNM. Transfected cells were cultured in media containing 10 µg/mL commercial HA. Three days after incubation, HA was purified and run on an agarose gel.

Supplementary Fig. S7B-E:

Figure legends: (B) Media containing 20 µg/mL commercial high molecular weight HA was incubated with growing fibroblasts from each species. Two days after incubation, HA was purified from the media and run on an agarose gel. For each cell line, a sample without commercial HA

was included to control the endogenous HA. The amount of loaded HA was normalized to cell number of each species. (C-E) 293T cells were transfected with either HYAL2 (C) or HYAL1 (D) of mouse, BMR, shrew, or SNM. Transfected cells were cultured in media containing 20 $\mu\text{g}/\text{mL}$ commercial HA. Two days after incubation, HA was purified and run on an agarose gel. (E) HA intensities were quantified using ImageJ. The relative degradation of HA by different HYAL2 was calculated in such a manner that the differences in HA intensity between tested group and commercial HA was normalized to the total intensity of commercial HA. Data are mean \pm SD of three technical replicates. * $P < 0.05$ by unpaired two-sided t-test.

- The authors go on to propose several mutations to HYAL2 that they suggest impacts their function but unfortunately do not test them. To publish in a high-impact journal such as this one I would expect further mechanistic study of the functional molecular differences that differentiate the regulation of HMM-HA in surface vs. subterranean species that goes well beyond PCR analysis.

Response: We generated five mutant plasmids corresponding to the five mutation sites of EM and SNM HYAL2. Using commercial HA incubation, we found that, when the SNM HYAL2 was mutated to the amino acid of shrew on each site, the HA-degrading ability became stronger in 293T cells (Supplementary Fig. S7F-G). When incubated for three days, mutants H106L, K284A, and A352V induced smaller sized HA compared with wild-type HYAL2 (Fig. 5h). These results suggest that at least the three unique mutations of EM and SNM HYAL2 resulted in a weaker HYAL2 activity, which contributed to the accumulation of HMM-HA. A positively selected site G284 was also identified in NMR and DMR HYAL2 (Fig. 3c). We tested the effect of this mutation on NMR HYAL2. We did not observe difference when HA was incubated for two days, but the size and amount of HA was smaller after three days of incubation (Fig. 5i), suggesting that this site mutation also contributed to the weaker HYAL2 of NMR.

Supplementary Fig. S7F-G:

Figure legends: (F) 293T cells were transfected with either wild-type (WT) SNM HYAL2 plasmid, or each of its five mutants. Cells were incubated with media containing 20 $\mu\text{g}/\text{mL}$ commercial HA for two days, followed by HA extraction and electrophoresis. (G) Quantification of HA intensities after degraded by WT or mutant SNM HYAL2. Data are mean \pm SD of three technical replicates. * $P < 0.05$ by unpaired two-sided t-test.

Fig. 5h-i:

Figure legends: Effects of SNM and NMR HYAL2 mutations on HA degradation. **h**, HeLa cells were transfected with either wild-type (WT) SNM HYAL2 plasmid, or each of its five mutants. **i**, HeLa cells were transfected with either NMR HYAL2 WT or the G284A mutant. Cells were incubated with media containing 10 μ g/mL commercial HA for three days, followed by HA extraction and electrophoresis.

- Statistical information is not provided anywhere. Please specify your methods of statistical analysis, including test and post-test information, alpha value, etc. Please provide f-stats where appropriate in the results section.

Response: We added statistical information at the end of Methods section. Information about replicates was provided in the results and figure legends.

Minor comments:

- The manuscript is generally well-written but there are numerous grammatical and syntax errors and I encourage the authors to proof-read their submission more carefully. Some examples (there are several others):

Response: We proof-read the whole manuscript and edited the grammatical errors.

o Abstract: authors use both HM weight and HM mass to describe HMM-HA

We changed 'weight' to 'mass' to keep consistency.

o ECM is defined 2x in the first intro paragraph

We removed the redundant definition of ECM in the paragraph.

o Lines 66 and 68: the ECM

Done.

o Line 67: delete comma

Done.

o Line 79: “superior...than” is not grammatically correct

We changed to “superior...to”.

- Abbreviations are used inconsistently: e.g., lines 111-112 contains long-form of some species and short form of others, even though abbreviations for all study species are introduced in the preceding paragraph. Same comment elsewhere, including in the methods.

Response: We edited all the abbreviations to keep consistency.

- Latin names are inconsistently provided (for all species except mouse and guinea pig)

Response: We provided Latin names for all species (except mouse and guinea pig) when they first appeared in the maintext and the Methods section.

- The authors refer to differences between gene expression in different species as being up or downregulated but this is somewhat misleading. These terms usually apply to changes in gene expression induced by a treatment, and not to differences between species at baseline. I suggest stating that some species had higher expression or higher endogenous expression instead.

Response: We changed all the descriptions to “higher/lower expression” in the text and figures.

- Lines 366-373: Not all subterranean species are hypoxia tolerant. Star-nosed moles are quite intolerant of hypoxic stress (see Devereaux et al, JEB 2021).

Response: We included this information in the discussion.

- Methods: provide animal numbers, sex, mass, and age.

Response: We added the information to the Methods section. It has to be noted that, however, the exact ages of wild-caught animals are difficult to determine. We used young adult according to their body size.

- Methods: Provide methods for obtaining fibroblast. Were animals anesthetized? Were they sacrificed? Please provide animal ethics approval information.

Response: The fibroblasts were isolated from sacrificed animals using the method established by our lab previously. We cited the paper describing the fibroblast isolation. Animal ethics approval was provided at the beginning of the Animals section.

- Methods: the body temperature of star-nosed moles is 35C

Response: We thank the reviewer for pointing this out. We cultured star-nosed mole cells at 37°C to keep it consistent with other cell lines, and for logistical reasons, and it did not cause stress or elevated cell death.

Matthew Pamerter

Reviewer #2 (Remarks to the Author):

This manuscript builds on previous studies from this group (ref 1) demonstrating that hyaluronan is increased in abundance and size in the naked mole rat as well as work from other groups (Ref 54 and Del Marmol et al [2021], Scientific Reports 11:7951) that examined the sequence and/or expression of genes involved in HA metabolism in the naked mole rat and other subterranean species compared to overland mammals.

Major Comments

1. This work expands the species demonstrated to have increased size and abundance of hyaluronan to include the Damaraland mole rat (DMR), Star-nosed mole (SNM), the Eastern mole (EM) and increased abundance to the Transcaucasian mole vole (TMV). This strengthens the hypothesis that increased abundance and sized hyaluronan is associated with a subterranean lifestyle and therefore would be of interest to groups working on the evolution of subterranean mammals as well as to researchers working in the hyaluronan field with interests in the metabolism of this extracellular molecule.

Response: We are grateful for the reviewer's appreciation to our paper's impact on subterranean evolution as well as the hyaluronan biology.

2. The authors extensively explore the gene expression profiles associated with a range of genes involved in HA metabolism as well as genes encoding protein constituents of the extracellular matrix. This is the first work to explore this broad range of genes in the context of multiple subterranean species. They also analyze the evolution and predicted functional impact of variants present in the hyaluronan-degrading enzymes HYAL1 and HYAL2, work that has not been done by any other groups. However, these studies could have been better described in the context of pre-existing work that did a thorough study of HAS2 evolutionary changes in 11 subterranean mammals and 57 other mammals (ref 54) as well as a study not referenced herein and mentioned above (Del Marmol et al) that provided additional evidence for elevated hyaluronan that was increased in size in the NMR (although not considered to have the high molecular mass suggested by this group) and showed that the only significant changes in hyaluronan-related genes in the NMR were increased expression of HYAL3 and TNFAIP6 genes.

Response: We added the description of the previous studies by ref 54 in the Introduction, and emphasized in the Discussion that our study extended the analysis to other HA-related enzymes beyond HAS2, especially HA-degrading enzymes including HYAL1 and HYAL2. Our results suggest that HAS2 and HYALs cooperate together to yield abundant HMM-HA in subterranean species. We also cited and described the study by Del Marmol et al in the Introduction emphasizing that at least for the NMR, other HA-related genes (including HYAL3 and TNFAIP6) also contribute to the HA size.

3. The authors claim that abundant HMM-HA is found in a wide variety of subterranean

mammalian species but not in phylogenetically related aboveground species. This argument could be strengthened by expanding the range of both subterranean and aboveground species that were examined. Compared to the study of HAS2 in moles (Ref 54) this study was relatively small.

Response: We expanded the species for comparison. We added two more subterranean species, the bamboo rat and plateau zokor. We also added more aboveground species from our previously published data (Lu JY, et al, 2022, Cell Metabolism). The results further supported our conclusion of the paper.

4. The second claim that these species accumulate abundant HMM-HA by regulating the expression of genes involved in HA degradation and synthesis and contain unique mutations in these genes is not strongly supported by the data in the manuscript. There were no consistent changes in the HAS2/HYAL2/HYAL1 genes of the subterranean mammals leading the authors to speculate that each species increases HA through a different mechanism. While this is possible, it is very speculative as HYAL1 works intracellularly and in mouse its complete loss can be completely compensated for by other exoglycosidases beta-glucuronidase and beta-hexosaminidase (Gushulak et al. [2012] 11:287:16689-97). Decreased HYAL2 or increased HAS2 activity are better candidates as are some of the other proteins, CEMIP and TMEM2. Overall the large variation in expression of these genes between biological replicates and between species made it difficult to come to any conclusion from these studies. The increased hyaluronan detected in the subterranean species may be completely unrelated to the expression of these genes and instead depend on the expression for example of a hyaluronan-binding protein.

Response: We agree that HAS2 and HYAL2 are the best indicators for HA. Our new results assessing the effects of HAS2 expression and HYAL2 mutations further supported the conclusion (see details in response to point 7, and Reviewer 1, Major comment 2). We analyzed the expression levels of HABP genes (mainly HABP2, which we used for staining, HABP4, and C1CBP), and they were not significantly different in expression levels across species (shown in figure below). Taken together, these results suggest that the more abundant HA we observed is the result of HAS2 and HYAL2 regulation. We included this information in the manuscript (included in Fig. 3b).

Fig. 3b:

Figure legend: Heatmap showing the expression fold changes for HA-related genes. Genes were grouped into six sets corresponding to their function, including HA substrate metabolic pathways (UDP-GlcNAc synthesis and UDP-GlcUA synthesis), HA synthesis, HA degradation, HA binding, and TNFAIP6 related genes. The fold change values were calculated by comparing each subterranean species to their closest aboveground species. For tissues that the closest aboveground species samples are not available, rat samples were used.

5. The analysis of hyaluronan levels in tissues (skin, heart, kidney) from the subterranean species using quantitative immunofluorescence was interesting but again would benefit from more structured analysis. Some of the things that made this less convincing were that the areas of the sections from within the tissues did not appear to be matched. The levels of hyaluronan vary considerably between different regions in tissues (see de Marmol ref and many others in the field). Therefore it is critical that regions of tissues are matched in the sections and that equivalent areas are quantified. These experiments would have benefitted by comparison to some kind of endogenous control that was considered to be unchanged. This data if carefully done could further substantiate the first claim that hyaluronan is increased in subterranean species.

Response: The immunofluorescence images showed representative samples. For quantification, similar regions of each organ were selected and the intensities were normalized to the area of selected regions. For endogenous control, we used Hoechst staining the nucleus to control the cell density. We added this information in the Methods.

6. The RNA seq analysis was an exciting addition to this study as it included 6 subterranean species and 4 aboveground species across 6 tissues. The inclusion of a broader range of genes involved in hyaluronan metabolism and function was great as HYAL3-6 are not known to have any role in hyaluronan metabolism. HYAL5 is a pseudogene, HYAL4 a chondroitinase, HYAL3 appears inactive, and HYAL6 is not present in all mammals. This work showed that HYAL3 was down-regulated in multiple tissues of subterranean species while the work of Marmol et al showed it to be upregulated up to 129 fold in NMR compared to mice. Given that the expression

levels were averaged for the aboveground and subterranean samples in this study it was difficult to try to reconcile this difference and the authors may want to address this directly. Further, are these studies consistent with the qPCR that was done in the first parts of the study on HAS2, HYAL1 and HYAL2 in specific tissues?

Response: We re-analyzed our RNA data and, instead of showing averaged data, we now show the expression of genes in tissues of individual species-pairs of subterranean versus most related above-ground species (Fig. 3b-see above). This analysis supports the previous finding by Marmol et al that HYAL3 is upregulated in NMR in multiple tissues, but also shows that HYAL3 is down-regulated in some other subterranean species. We also analyzed the expression of HAS2, HYAL1, and HYAL3 from the RNAseq data, and they were generally consistent with our qPCR results. We included this result in the manuscript (Supplementary Fig. S5B).

Supplementary Fig. S5B:

Figure legend: (B) Boxplots showing the expression of HAS2, HAYL1, and HYAL2 in skin, kidney and heart of different species determined by RNAseq. Log2 transformed normalized counts were shown.

7. The analysis of variants in HYAL1 and HYAL2 was interesting and the authors did strive to predict the impact of these changes by analyzing how these changes would affect the structure of HYAL2 in a model of HYAL2 generated with AlphaFold. These studies would need to be followed up by some in vitro assays of HYAI2 harboring these variants. The size of hyaluronan in the medium of cells transfected with wild type vs these variant-containing forms of HYAL2 would help to provide additional insight into the significance of these variants in the context of HYAL2 activity.

Response: We did a thorough research by assessing the effects of HYLAs on HA degradation in the subterranean species. The new figures were included and the figure numbers were highlighted:

We first compared the HA degradation abilities between subterranean and aboveground species. Commercial high molecular mass HA was incubated with confluent fibroblasts of five subterranean species or three aboveground control species for two days. After incubation, more HA remained after incubated with cells of subterranean species than aboveground species (Fig. 4c). The result remained the same when the HA was incubated with growing cells (Supplementary Fig. S7B), suggesting that subterranean species' cells had weaker HA-degrading abilities. We then examined the different HA-degrading abilities of HYAL2 and HYAL1 in subterranean species. We overexpressed either HYAL2 or HYAL1 of mouse, BMR, shrew, or SNM in 293T cells. Commercial HA was incubated with the transfected cells for two days, and HA was extracted and subjected to electrophoresis. The shrew HYAL2 induced more degradation to commercial HA than the SNM HYAL2 (Supplementary Fig. S7C-D), suggesting that the HYAL2 in subterranean species had weaker HA-degrading abilities. We performed the same experiment in HeLa cells. While no differences were observed after two days incubation, when the incubation was extended to three days, significant amount of commercial HA was degraded. Strikingly, mouse and shrew HYAL2 degraded more HA, resulting in smaller size of remaining HA compared with BMR and SNM, respectively (Fig. 4d). As expected, cells expressing HYAL1 of different species did not yield significant difference in HA degradation (Fig. 4d and Supplementary Fig. S7E), probably because HYAL1 is an intracellular enzyme. Taken together, these results suggest that subterranean HYAL2 enzymes had weaker HA-degrading activities, contributing to HMM-HA.

Fig. 4c-d:

Figure legends: **c**, Media containing 20 $\mu\text{g}/\text{mL}$ commercial high molecular weight HA was incubated with confluent fibroblasts from each species. Two days after incubation, HA was purified from the media and run on an agarose gel. For each cell line, a sample without commercial HA was included to control the endogenous HA. **d**, HeLa cells were transfected with either HYAL2 or HYAL1 of mouse, BMR, shrew, or SNM. Transfected cells were cultured in media containing 10 $\mu\text{g}/\text{mL}$ commercial HA. Three days after incubation, HA was purified and run on an agarose gel.

Supplementary Fig. S7B-E:

Figure legends: (B) Media containing 20 µg/mL commercial high molecular weight HA was incubated with growing fibroblasts from each species. Two days after incubation, HA was purified from the media and run on an agarose gel. For each cell line, a sample without commercial HA was included to control the endogenous HA. The amount of loaded HA was normalized to cell number of each species. (C-E) 293T cells were transfected with either HYAL2 (C) or HYAL1 (D) of mouse, BMR, shrew, or SNM. Transfected cells were cultured in media containing 20 µg/mL commercial HA. Two days after incubation, HA was purified and run on an agarose gel. (D) HA intensities were quantified using ImageJ. The relative degradation of HA by different HYAL2 was calculated in such a manner that the differences in HA intensity between tested group and commercial HA was normalized to the total intensity of commercial HA. Data are mean ± SD of three technical replicates. * P < 0.05 by unpaired two-sided t-test.

We then generated five mutant plasmids corresponding to the five mutation sites of EM and SNM HYAL2. Using commercial HA incubation, we found that, when the SNM HYAL2 was mutated to the amino acid of shrew on each site, the HA-degrading ability became stronger in 293T cells (Supplementary Fig. S7F-G). When incubated for three days, mutants H106L, K284A, and A352V induced smaller sized HA compared with wild-type HYAL2 (Fig. 5h). These results suggest that at least the three unique mutations of EM and SNM HYAL2 resulted in a weaker HYAL2 activity, which contributed to the accumulation of HMM-HA. A positively selected site G284 was also identified in NMR and DMR HYAL2 (Fig. 3c). We tested the effect of this mutation on NMR HYAL2. We did not observe difference when HA was incubated for two days, but the size and amount of HA was smaller after three days of incubation (Fig. 5i), suggesting that this site mutation also contributed to the weaker HYAL2 of NMR.

Supplementary Fig. S7F-G:

Figure legends: (F) 293T cells were transfected with either wild-type (WT) SNM HYAL2 plasmid, or each of its five mutants. Cells were incubated with media containing 20 $\mu\text{g}/\text{mL}$ commercial HA for two days, followed by HA extraction and electrophoresis. (G) Quantification of HA intensities after degraded by WT or mutant SNM HYAL2. Data are mean \pm SD of three technical replicates. * $P < 0.05$ by unpaired two-sided t-test.

Fig. 5h-i:

Figure legends: Effects of SNM and NMR HYAL2 mutations on HA degradation. **h**, HeLa cells were transfected with either wild-type (WT) SNM HYAL2 plasmid, or each of its five mutants. **i**, HeLa cells were transfected with either NMR HYAL2 WT or the G284A mutant. Cells were incubated with media containing 10 $\mu\text{g}/\text{mL}$ commercial HA for three days, followed by HA extraction and electrophoresis.

We added these new results in the manuscript.

8. The last section of the paper looking at ECM related genes is extremely interesting but felt disconnected from the focus of the paper. It does make sense to go this direction given the lack of a clear linkage with HA metabolic enzymes, but the authors could improve the linkages. This section of the paper seemed to move more in the direction of looking at ECM components as a

function of evolution of subterranean species. One apparent link comes from the study of Del Marmol where the TSG-encoding gene is differentially expressed in the NMR compared to the guinea pig. Was this finding reproduced in these studies?

Response: Our analysis of ECM related genes showed co-evolution of HA and ECM. We analyzed several TSG-encoding genes (Fig. 3b), and they showed higher expression (especially TNFAIP6) in NMR, which reproduced the result by Del Marmol et al.

9. The methods sections were detailed enough to allow the experiments to be reproduced.

Response: We thank the reviewer for approving our Methods section.

Minor Comments:

1. The authors sometimes missed the “d” on the star-nosed mole

Response: We changed all “star-nose mole” to “star-nosed mole” throughout the manuscript.

2. On line 82, the authors describe the size of the NMR HA as extensive; perhaps this should be large rather than extensive which has a different meaning.

Response: Agreed, we changed it to “large”.

3. The authors should keep in mind that some of these genes are affected by the confluency of the fibroblasts (HYAL1 for example, BBRC 266:268-273)

Response: This is a good point! We noticed the influences of cell confluency on gene expression and HA synthesis; therefore, we kept cells confluent for 10 days and then harvested the cells for qPCR. This is the same timepoint when the conditioned media was collected for HA extraction. We added this information in the Methods section.

4. The authors did not comment on the limitations of the carbazole assay which will also react with sugars present in the tissue or derived from other polysaccharides. They might consider using an assay specific to hyaluronan such as an HA ELISA (<https://www.echelon-inc.com/product/hyaluronan-enzyme-linked-immunosorbent-assay-ha-elisa/> [https://urldefense.com/v3/https://www.echelon-inc.com/product/hyaluronan-enzyme-linked-immunosorbent-assay-ha-elisa/ ;!!CGUSO5OYRnA7CQ!c7F75IsBGNxFAlE5hHEAwZSfQHNukoGWGzCwvp1OPjU6-BHPQY77R1mGv7L6foB8quwVQv4LqSM1zsKQntI3z4NV7T0wx0rc1Jve\\$>](https://urldefense.com/v3/https://www.echelon-inc.com/product/hyaluronan-enzyme-linked-immunosorbent-assay-ha-elisa/))

Response: We tried HA ELISA and it showed a similar trend between mouse and BMR (figure below, left panel). However, due to the requirement of dilution, the results varied between samples within groups, we therefore could not get conclusive results for other species. Alternatively, we quantified the intensity on the HA gel, which showed a similar trend (figure below, right panel), supporting our Carbazole results. We added this information in the paper (Supplementary Fig. S1).

a

b

Figure legend: Quantification of HA. HA quantified by ELISA (a) or gel intensity (b) showed similar trends between mouse and BMR.

5. The authors should indicate how they normalized the qPCR for RNA levels in these various species.

Response: The qPCR was normalized to β -actin. We added this information in the Methods section.

6. The authors should indicate if all known hyaluronan binding proteins were included in their ECM related genes.

Response: We included all known HA binding proteins for the analysis of ECM related genes. We added this information in the manuscript (Fig. 3b).

7. The authors do not indicate the sex of the animals that were used in these studies or describe any known sex-specific differences in hyaluronan tissue levels.

Response: We did not observe sex-dependent differences in the NMR. For other wild-caught animals, young adults with random sexes were obtained. We included this information in the manuscript.

Reviewer #3 (Remarks to the Author):

In the manuscript, Zhao et al. reported that the subterranean mammalian species from multiple independent lineages generally contained abundant high-molecular-mass hyaluronic acid (HMM-HA) which has been proven to contribute to the cancer resistance of the naked mole-rat (Tian et al. 2013, Nature), suggesting a general role of HMM-HA in adapting to the underground lifestyle. The authors next performed several evolutionary analyses on the genes involved in HA degradation and synthesis, and found strong signals of the adaptation in these genes, further suggesting the abundant HMM-HA may benefit the adaptation to subterranean environment.

Overall, the results are very inspiring and would be very interesting to the relevant field. Therefore, I would be happy to recommend acceptance after the authors make some revisions.

Specific comments:

1) The authors raised the scientific question that “However, the role of extracellular matrix (ECM) in adaptation to subterranean environment, remains poorly understood” at the end of the

first paragraph of Introduction. However, this is not the exact question that the study wanted to address. The authors should clarify the question about the evolution of HMM-HA in adaptation to the subterranean lifestyle of mammals.

Response: We thank the reviewer for pointing this out. Yes, the focus is HMM-HA in subterranean adaptation. We revised the statement in the Introduction.

2) The authors missed the results of evolutionary analyses about the genes involved in HMM-HA degradation and synthesis at the end of Introduction. Please add the related information.

Response: We added the description of the results of HMM-HA degradation and synthesis genes at the end of Introduction.

3) In the first part of Results, the authors should display the statistical results in the maintext, just like in the following parts about evolutionary analyses. This is also true for the RT-qPCR results. In the second paragraph, they only compared the size of HA, not the amount of HA, please modify the first sentence of this paragraph.

Response: We added the statistical results in the main text of Results section. The exact p-values between each group are available in the original data deposited. We modified the first sentence of the second paragraph.

4) The authors missed the methods about how to test the relaxed selection and how to obtain the sequences. Please add the information.

Response: We added this information in the Methods section.

5) When displaying the parallel amino acid replacements (Fig. 3d), it would be better to show these sites according to the species' phylogeny.

Response: We edited Fig. 3d to show these sites according to the species' phylogeny.

6) In fact, besides the subterranean mammals the authors mentioned in the study, there were several additional underground mammalian species with available high-quality genome sequences such as bamboo rat (Guo et al. 2021; doi.org/10.24272/j.issn.2095-8137.2021.240<<http://doi.org/10.24272/j.issn.2095-8137.2021.240>>) and plateau zokor (Xu et al. 2021; doi.org/10.1038/s41422-021-00517-6<<http://doi.org/10.1038/s41422-021-00517-6>>). The authors are suggested to include these species in their evolutionary analyses to improve the generality of their main point that the evolution of high-molecular-mass hyaluronic acid is associated with the subterranean lifestyle of mammals.

Response: We thank the reviewer for this suggestion. We added bamboo rat and plateau zokor to the analysis (Fig. 6b and Supplementary Fig. S8B). The results support our original conclusion. We updated the results in the manuscript.

Fig. 6b:

Figure legend: Phylogenetic tree of species in this study and their numbers of positively selected genes (PSGs), convergently evolving genes (CEGs), convergent-to-divergent site ratio (*C/D* ratio; > 3 species) and genes with both positive selection sites and convergent sites (CEGs-PSGs). Error bars indicates the standard deviation of numbers of CEGs and *C/D* ratio resulted from different species combinations. Subterranean species are shown in red.

Supplementary Fig. S8B:

Figure legend: Alignment of NID1 protein sequence. Locations of two convergent mutation sites of NID1 protein across subterranean species were indicated with arrows. The names of subterranean species were labeled in red.

REVIEWER COMMENTS

Reviewer #1 (Remarks to the Author):

Thank you for your robust response to my concerns and questions. The additional data has greatly strengthened this study and I have enjoyed peer-reviewing this work. I have no further concerns.

Matthew Pamerter

Reviewer #2 (Remarks to the Author):

The authors have done a number of experiments since the last submission. The analysis of HA is not easy and unfortunately, I do not feel that the authors have advanced their case toward demonstrating that regulation of HAS2, HYAL1 and HYAL2 contribute to the increased HMM-HA in subterranean species. Their conclusion "Further analysis revealed that the differential expression of HAS2, HYAL1, and HYAL2, as well as five unique mutations on HYAL2 contribute to the accumulation of abundant HMM-HA in the subterranean species." is overstated.

Related to Previous Comment #3 to the authors:

I cannot find the results of these additional subterranean species in their analysis of hyaluronan viscosity or size (Figure 1b and 1c). In supplementary Fig. S1a looking at HA in the media using an HA ELISA there does not appear to be any significant difference between BMR and mouse. The data from HA on a gel (S1b) is a single bar with no indicators that it is more than a single assay.

Related to Previous Comment #4

4. The authors have performed a number of RNA analyses which still leave the mode of regulation of HA levels unclear. The reviewer appreciates the many experiments performed by the authors which show some apparent differences in the level of HA in subterranean vs above ground species (supp Fig4c-d), but the approach to these experiments still does not differentiate the role of HAS2, HYAL2, etc. For example, in the media from above ground species, the levels appear similar to that of the control lane (R & D HA) but increased in the subterranean species suggesting to me increased synthesis rather than decreased degradation, or perhaps an altered internalization. I do not think that these experiments help to address regulation.

Related to Previous Comment #5

The types of structures in the skin sections, and direction of the sections, do not appear to be well matched. HA levels change dramatically around different types of structures and therefore matching the types of structure in a section is extremely important. Are you looking at a hair follicle for example or sebaceous gland. It does not make sense to take average intensity if the tissue contains different structures. Also, in mouse vs TMV in S2a, the level of HA appears higher in mouse than TMV because even though the intensity is not greater, the % of volume covered by the HA is greater. Is mean intensity a suitable measure of HA? Volume will have to also be accounted.

Related to Previous Comment #6

The data was separated out for individual species and the authors indicate it is largely consistent with that reported in the first parts of the study. No detailed discussion was provided.

Related to Previous Comment #7 and other comments above:

The applicant did a great deal of work to characterize mutant forms of HYAL2 and regulation of HA-metabolizing enzymes that resulted in several new figures. The major problem with these experiments is that there was no control of expression levels so that different transfection efficiencies (Fig 4D, Fig. S7C-E) and different levels of expression of mutant proteins (Fig.S7F-G, Fig. 5h-i) were not taken into

account in their analyses. At one point the authors indicate that they have a 14 fold higher expression but there is no western blot to demonstrate this. Further, the addition of exogenous HA which may not function in the same way as endogenous HA that is bound with protein. As a result many of the experiments that were performed are not well enough controlled to be interpretable or convincing that it reflects endogenous function. Based on these experiments it is difficult to make conclusions about the regulation of HAS2, HYAL2 and HYAL2.

Many of the small comments from this reviewer are addressed.

Reviewer #3 (Remarks to the Author):

The authors have fully addressed my questions. I have no more concerns.

REVIEWER COMMENTS

Reviewer #1 (Remarks to the Author):

Thank you for your robust response to my concerns and questions. The additional data has greatly strengthened this study and I have enjoyed peer-reviewing this work. I have no further concerns.

Matthew Pamenter

Response: We thank the reviewer for approving our manuscript and for their constructive suggestions that helped to improve our paper.

Reviewer #2 (Remarks to the Author):

The authors have done a number of experiments since the last submission. The analysis of HA is not easy and unfortunately, I do not feel that the authors have advanced their case toward demonstrating that regulation of HAS2, HYAL1 and HYAL2 contribute to the increased HMM-HA in subterranean species. Their conclusion “Further analysis revealed that the differential expression of HAS2, HYAL1, and HYAL2, as well as five unique mutations on HYAL2 contribute to the accumulation of abundant HMM-HA in the subterranean species.” is overstated.

Related to Previous Comment #3 to the authors:

I cannot find the results of these additional subterranean species in their analysis of hyaluronan viscosity or size (Figure 1b and 1c). In supplementary Fig. S1a looking at HA in the media using an HA ELISA there does not appear to be any significant difference between BMR and mouse. The data from HA on a gel (S1b) is a single bar with no indicators that it is more than a single assay.

Response: The cells of these additional species are logistically unavailable. Therefore, we were not able to analyze their viscosity or HA size. Instead, we included published sequences of these species in our evolutionary analysis using bioinformatics (Figure 3e, Figure 6, Supplementary Figure S9b and S10), which strengthened our conclusions.

We repeated the HA ELISA comparing BMR and mouse with more replicates. The result was significant and we updated this figure (Fig. S1a, see below).

For HA gel, we quantified three HA gels from three independent experiments and showed averaged intensities with error bars. The differences between species are significant as indicated in the figure (Fig. S1b, see below).

Figure legend: **Quantification of HA in conditioned media.** (a) ELISA quantification of HA from media of BMR and mouse cells. Data are mean \pm SD of biological replicates. *** $P < 0.001$ by unpaired two-sided t-test. (b) Quantification of HA intensity from pulsed-field gel. HA purified from media of different cell lines was subjected to pulsed-field electrophoresis and stained with Stains-All. The relative HA intensity was quantified using ImageJ. Data are mean \pm SD of gels from three independent experiments. * $P < 0.05$ by unpaired two-sided t-test.

Related to Previous Comment #4

4. The authors have performed a number of RNA analyses which still leave the mode of regulation of HA levels unclear. The reviewer appreciates the many experiments performed by the authors which show some apparent differences in the level of HA in subterranean vs above ground species (supp Fig4c-d), but the approach to these experiments still does not differentiate the role of HAS2, HYAL2, etc. For example, in the media from above ground species, the levels appear similar to that of the control lane (R & D HA) but increased in the subterranean species suggesting to me increased synthesis rather than decreased degradation, or perhaps an altered internalization. I do not think that these experiments help to address regulation.

Response: In the experiment in Fig. 4c, both media with or without R&D HA were included to show that subterranean species degrade less HA. Only the left lane of each species (with "HA-") indicates the synthesis of HA by the cells. Also, in Supp Fig. 7b, we used growing cells, which did not give them time to synthesize much endogenous HA, and the result supported that subterranean species have weaker HA degradation (more R&D HA left on the gel). Collectively, our results (Fig. 4) suggest that both synthesis and degradation contribute to HMM-HA in subterranean species, and the relative contribution differs by species. Therefore, it is expected that some subterranean species show higher synthesis in Fig. 4c. For example, BMR HAS2, with high endogenous expression level, synthesized more HA than mouse HAS2 (Fig. 4a). This can explain the more abundant HA in the gel of Fig. 4c (compare lane 10 vs. lane 4).

Related to Previous Comment #5

The types of structures in the skin sections, and direction of the sections, do not appear to be well matched. HA levels change dramatically around different types of structures and therefore matching the types of structure in a section is extremely important. Are you looking at a hair follicle for example or sebaceous gland. It does not make sense to take average intensity if the tissue contains different structures. Also, in mouse vs TMV in S2a, the level of HA appears higher in mouse than TMV because even though the intensity is not greater, the % of volume covered by the HA is greater. Is mean intensity a suitable measure of HA? Volume will have to also be accounted.

Response: To address this concern we re-quantified the HA in well-matched regions of the comparable samples. It is difficult to avoid hair follicles in the skin sections, but we selected regions with similar tissue structures outside the hair follicles for each sample and re-quantified them. Mean intensity is routinely used to determine abundance in immunofluorescence. The mean intensity was calculated by the cumulative intensity normalized by the area (volume) of analysis. Therefore, our quantification already took volume into consideration. The results support our conclusion that subterranean species have more abundant HA in the similar regions of skin. We added a square with dashed outline to indicate the regions used for quantification. Other replicates were analyzed in the same manner and the figure only shows one representative section for each species.

We re-analyzed Fig. S2a for mouse and TMV, and HA levels in mouse heart were slightly higher than in TMV but the difference was not significant. We updated these results (Figures below).

Fig. 2a:

Figure legends: **Skin of subterranean species contains higher levels of HA.** **a**, Cryosections of skin of different species were stained with hyaluronan binding protein (HABP, red) and Hoechst (blue). Representative immunofluorescence images are shown at 20 × magnification (left). Scale bar, 50 μm. Quantification of HABP fluorescence intensity (right panel) was performed on the similar regions of each sample as indicated by the squares in dashed line (left panel). Data are mean ± SD of three independent areas. * $P < 0.05$, ** $P < 0.01$, and *** $P < 0.001$ comparing subterranean species with their aboveground controls by unpaired two-sided t-test.

Supplementary Fig. S2a:

Figure legends: **Levels of HA in the heart of subterranean species.** (a) Cryosections of heart tissues from different species were stained with hyaluronan binding protein (HABP, red) and Hoechst (blue). Experiments were repeated three times. Representative Immunofluorescence images are shown at 20 × magnification (left). Scale bar, 50 μm. Quantification of HABP fluorescence (right). Data are mean ± SD of three independent areas. *P*-value indicates the difference between subterranean and the corresponding aboveground species. * *P* < 0.05 by unpaired two-sided t-test.

Related to Previous Comment #6

The data was separated out for individual species and the authors indicate it is largely consistent with that reported in the first parts of the study. No detailed discussion was provided.

Response: We provided more detailed discussion in the text comparing the qPCR and RNAseq data: “Particularly, EM and SNM HAS2 were more highly expressed in the skin, heart, and kidney compared to the shrew as demonstrated in both qPCR and RNAseq data (Fig. 2b, Supplementary Fig. S2b, S3b, and S5b). The expression levels of HAS2 were higher in the skin of NMR and DMR than in GP (Fig. 2b, Supplementary S5b). The expression levels of HYAL1 were significantly lower in the heart and kidney of NMR than in GP (Supplementary Fig. S2b, S3b, and S5b).”

We highlighted this discussion in the text.

Related to Previous Comment #7 and other comments above:

The applicant did a great deal of work to characterize mutant forms of HYAL2 and regulation of HA-metabolizing enzymes that resulted in several new figures. The major problem with these experiments is that there was no control of expression levels so that different transfection efficiencies (Fig 4D, Fig. S7C-E) and different levels of expression of mutant proteins (Fig.S7F-G, Fig. 5h-i) were not taken into account in their analyses. At one point the authors indicate that they have a 14 fold higher expression but there is no western blot to demonstrate this. Further, the addition of exogenous HA which may not function in the same way as endogenous HA that is bound with protein. As a result many of the experiments that were performed are not well enough controlled to be interpretable or convincing that it reflects endogenous function. Based on these experiments it is difficult to make conclusions about the regulation of HAS2, HYAL2 and HYAL2.

Response: To control for transfection efficiency, we quantified the expression of the transfected genes. Because of the sequence variations across different species, Western blot would not provide reliable quantification due to differences in antibody affinity. Furthermore, the sequence variations of these proteins may influence protein stability as we predicted by structural analysis (Fig. 5). We therefore used qPCR to determine the transfection efficiency. The results show that BMR HAS2 had significantly

higher expression levels than mouse HAS2, mimicking the differences in endogenous expression level. All the other corresponding genes had similar expression levels (Fig. S7). Modest differences between mutant proteins were observed due to high sensitivity of qPCR. These results confirm that the transfection efficiency and mRNA expression levels were similar across groups, which further validates our conclusions. We included these results in the new Supplementary Fig. S7. We also added discussion about transfection efficiency (highlighted in the text).

Supplementary Fig. S7:

Figure legends: **Expression levels of transfected genes.** HeLa cells (a) and 293T cells (b) were transfected with low amount of mouse HAS2 or high amount of BMR HAS2 to mimic the different endogenous expression in skin fibroblasts. Other genes including HYAL1 and HYAL2 from mouse, BMR, shrew, SNM, as well as mutant HYAL2 of SNM and NMR were transfected equally. RT-qPCR

was performed to determine expression levels of HAS2, HYAL2, and HYAL1 in transfected cells. Data are mean \pm SD of three technical replicates. *** $P < 0.001$ by unpaired two-sided t-test.

Many of the small comments from this reviewer are addressed.

Response: We thank the reviewer for helping us improve our manuscript.

Reviewer #3 (Remarks to the Author):

The authors have fully addressed my questions. I have no more concerns.

Response: We thank the reviewer for approving our manuscript and for their constructive suggestions that helped to improve our paper.

REVIEWERS' COMMENTS

Reviewer #2 (Remarks to the Author):

Thank-you for addressing my concerns. I believe the study has been strengthened by these additional experiments/analyses. I have no further concerns.